



**Title: Nitrification and ammonium dynamics in Lake Taihu, China: seasonal competition**
**for ammonium between nitrifiers and cyanobacteria.**
Justyna J. Hampel[1*], Mark J. McCarthy[1,2], Wayne S. Gardner[2], Lu Zhang[3], Hai Xu[3], Guangwei
Zhu[3], Silvia E. Newell[1]
[1] Department of Earth & Environmental Sciences, Wright State University, Dayton, OH
[2]The University of Texas at Austin, Marine Science Institute, Port Aransas, TX
[3]Taihu Laboratory for Lake Ecosystem Research, Nanjing Institute of Geography and
Limnology, Chinese Academy of Sciences, Nanjing, China
[*] Corresponding author (hampel.4@wright.edu)



**Abstract**
Taihu Lake is hypereutrophic and experiences seasonal, cyanobacterial harmful algal blooms.
These *Microcystis* blooms produce microcystin, a potent liver toxin, and are linked to
anthropogenic nitrogen (N) and phosphorus (P) loads to lakes. *Microcystis spp.* cannot fix
atmospheric N and must compete with ammonia-oxidizing and other organisms for ammonium
($NH_4^+$). We measured $NH_4^+$ regeneration and potential uptake rates and total nitrification using
stable isotope techniques. Nitrification studies included separate $NH_4^+$ and nitrite ($NO_2^-$)
oxidation rates and abundance of the functional gene for $NH_4^+$ oxidation, *amoA*, for ammonia-
oxidizing archaea (AOA) and bacteria (AOB). Potential $NH_4^+$ uptake rates ranged from 0.02–
6.80 µmol $L^{-1}$ $hr^{-1}$ in the light and 0.05–3.33 µmol $L^{-1}$ $hr^{-1}$ in the dark, and $NH_4^+$ regeneration
rates ranged from 0.03–2.37 µmol $L^{-1}$ $hr^{-1}$. Nitrification rates exceeded previously reported rates
in most freshwater systems. Total nitrification often exceeded 200 nmol $L^{-1}$ $d^{-1}$ and exceeded
1000 nmol $L^{-1}$ $d^{-1}$ at one station near a river discharge. In Meiliang Bay and the open lake,
average $NO_2^-$ oxidation rates (248 ± 39.0 nmol $L^{-1}$ $d^{-1}$) exceeded $NH_4^+$ oxidation rates (22.0 ±
6.00 nmol $L^{-1}$ $d^{-1}$; $p < 0.001$) by an order of magnitude across all sampling events. AOA *amoA*
gene copies were more abundant than AOB gene copies ($p < 0.005$) at all times; however, only
abundance of AOB *amoA* (not AOA) was correlated with nitrification rates for all stations and
all seasons ($p < 0.005$). Regeneration results suggested that cyanobacteria relied extensively on
regenerated $NH_4^+$ to sustain the bloom in late summer. Nitrification rates in Taihu varied
seasonally; at most stations, rates were highest in March, lower in June, and lowest in July,
corresponding with cyanobacterial bloom progression, suggesting that nitrifiers are poor
competitors for $NH_4^+$ during the bloom. Internal $NH_4^+$ regeneration exceeded external N loading
to the lake by a factor of two and is ultimately fueled by external N loads. Our results thus



support the growing literature calling for watershed N loading reductions in concert with existing
management of P loads.



## 1. Introduction

Nitrogen (N) and phosphorus (P) are important nutrients in aquatic ecosystems, often co-
limiting primary production (Elser et al., 2007). Biologically unavailable (except to diazotrophs)
atmospheric N can be fixed to readily assimilable ammonium ($NH_4^+$) and biomass via N fixation
(Vitousek et al., 2013). However, fertilizer production from anthropogenic N fixation (the Haber-
Bosch process) has changed N cycling and the global N budget over the last century. Non-point
source N loads from agriculture are a main driver of eutrophication in aquatic systems, which is
often manifested as hypoxia, loss of biodiversity, cyanobacterial harmful algal blooms
(cyanoHABs; Paerl et al., 2016; Paerl and Paul, 2012), and other detrimental characteristics.
CyanoHABs are particularly problematic because they often produce toxins, compete for
nutrients with other microbes and primary producers, and indicate unhealthy aquatic systems.
The increase in extent and frequency of cyanoHABs correlates to increased application of
$NH_4^+$ and urea fertilizers, both globally and in China (Glibert et al., 2014). Diatoms are
competitive for oxidized forms of N (e.g., $NO_3^-$), but non-$N_2$ fixing cyanobacteria, such as
*Microcystis*, thrive on chemically reduced N forms, such as $NH_4^+$ and urea (Blomqvist et al.
1994; Glibert et al., 2016; McCarthy et al., 2009). $NH_4^+$ transport across the cell membrane and
assimilation into biomass is less energy intensive than for $NO_3^-$ (Glibert et al., 2016). Due to high
biological demand and fast turnover rates, $NH_4^+$ often does not accumulate in the water column,
resulting in low *in situ* concentrations. Ammonium regeneration is especially important to
phytoplankton productivity in productive eutrophic systems (Gardner et al. 1998, 2017;
McCarthy et al., 2013). For example, water column regeneration was up to six times higher than
sediment regeneration in Lake Taihu, China (McCarthy et al., 2007; Paerl et al., 2011).



Nitrification is the link between chemically reduced and oxidized N forms. Most
nitrification pathways are a two-step process; $NH_4^+$ is oxidized to nitrite ($NO_2^-$) via ammonia
oxidation, and $NO_2^-$ is then oxidized to $NO_3^-$ via $NO_2^-$ oxidation. Ammonia oxidation is a rate
limiting step (Ward, 2008) carried out by chemolithoautotrophic, ammonia oxidizing bacteria
(AOB) and ammonia oxidizing archaea (AOA; Könneke et al., 2005). $NO_2^-$ oxidation is carried
out by $NO_2^-$ oxidizing bacteria (NOB). Recently, a species of NOB was described that is capable
of one step, complete nitrification ("comammox"); however, comammox bacteria have yet to be
well documented in the environment (Daims et al., 2015). The ammonia and $NO_2^-$ oxidation
steps are often tightly coupled, where the product of the first step serves as a substrate for the
second step (Ward, 2008). However, some studies in marine environments suggest that the
process can be decoupled, with one step outpacing the other (Füssel et al., 2012; Heiss and
Fulweiler, 2016).
In Taihu, the abundance of ammonia oxidizing organisms (AOO) was investigated in
sediments, where AOA outnumbered AOB, often by an order of magnitude (Wu et al., 2013;
Zeng et al., 2012; Zhao et al., 2013). Another sediment study revealed that, while AOO were
present at all sites, the distribution of AOA and AOB depended on lake trophic status (Hou et al.,
2013). Abundance of AOA decreased, while AOB increased, with increasing trophic status,
following the substrate concentration hypothesis presented in kinetic experiments (Martens-
Habbena et al., 2009). A suite of environmental variables (substrate concentration, oxygen
concentration, light intensity, pH, etc.) influences nitrification rates and AOO community
composition, including AOA and AOB relative abundances (Bristow et al., 2015; Merbt et al.,
2012; Ward, 2008)



Nitrification can be closely coupled in time and space to N removal via denitrification,
particularly in shallow systems with tightly coupled benthic-pelagic interactions (An and Joye,
2001; Jenkins and Kemp, 1984). Microbial removal of excess N in eutrophic systems is a crucial
process to mitigate excessive N loads, and substrate availability for denitrification can depend on
nitrification. However, nitrifiers must compete with phytoplankton and other primary producers
for $NH_4^+$. In eutrophic systems, this competition could help determine microbial community
structure and cyanoHAB severity. Although both AOO and cyanoHABs, such as *Microcystis*,
have a strong affinity for $NH_4^+$, we are unaware of measurements made when AOO and
cyanoHABs were in direct competition. At some point in the bloom progression, cyanoHABs
must outcompete AOO for available $NH_4^+$.
The overall objective of this study was to investigate seasonal $NH_4^+$ dynamics and the
degree of competition between AOO and cyanoHABs in hypereutrophic Taihu. We measured
$NH_4^+$ regeneration, nitrification, and potential uptake rates under different bloom conditions to
help determine how cyanoHABs influence $NH_4^+$ fluxes. We compare these rates to: (1)
investigate the competition for $NH_4^+$ between phytoplankton/cyanobacteria and nitrifying
bacteria and archaea; (2) quantify the oxidation of $NH_4^+$ to $NO_3^-$, which is in turn available for
removal via denitrification or assimilation by other organisms; (3) determine the fraction of
$NH_4^+$ that is supplied internally via water column regeneration/remineralization; and (4)
characterize the community composition of AOO. We hypothesized that: (1) lower nitrification
rates occur during cyanoHABs due to increased competition for $NH_4^+$; (2) rates of nitrification
are greater in Taihu than in most coastal and marine systems due to high *in situ* substrate
concentrations; (3) rapid $NH_4^+$ turnover increases with phytoplankton biomass; and (4) AOB
outnumber AOA due to higher saturation concentrations.



## 2. Methods

### 2.1 Site description and time frame

Lake Tai (Taihu; from the Chinese for "Great Lake") is China's third largest freshwater lake. Due to industrial development and urbanization in the watershed, Taihu has shifted from a diatom-dominated, mesotrophic lake to a hypereutrophic lake experiencing cyanoHABs (Paerl et al., 2014; Qin et al., 2007). These blooms are associated with toxin producing, non-$N_2$ fixing *Microcystis spp.,* which can form surface scums on the lake for up to 10 months per year (Duan et al., 2009; Ma et al., 2016). The surface blooms have a well-documented negative impact on fisheries, tourism, and local economies, including a drinking water shutdown in 2007 (Qin et al., 2007; Steffen et al., 2017; Xu et al., 2010).

Taihu is a large (2,338 $km^2$), shallow (mean depth = 1.9 m) lake in southeast China, situated in the Yangtze river delta about 150 km west of Shanghai. The lake is an important source of freshwater and resources for the ~40 million people within the watershed. Taihu has a complicated hydrology, with 172 rivers and channels connected to the lake (Qin et al., 2007). This network of rivers carries nutrient loads from agricultural runoff, factories, and household wastewater.

Water samples were collected from four locations: Stations 1 and 3 in Meiliang Bay, Station 7 in the north-central part of the lake, and Station 10 on the western side of the lake basin (Fig. 1). In previous studies (e.g., McCarthy et al., 2007), sampling Stations 1, 3, and 7 followed a discharge gradient from the Liangxihe River in the northeast part of Meiliang Bay to the central lake, and Station 0 ("river") was located at the Liangxihe River discharge. However, in 2007, the Yangtze River was diverted into Taihu in an effort to decrease the lake residence time and flush *Microcystis spp.* and nutrients out of the lake (Qin et al., 2010). Diverted water from the Yangtze



River now flows into Gonghu Bay, the easternmost of the three northern bays. This diversion
resulted in intermittent flow reversals through Meiliang Bay, where the Liangxihe River now
mainly serves as an outflow. Since the discharge gradient from Station 1 to 7 was no longer
consistent in Meiliang Bay, Station 0 was replaced with a new river input (Station 10) on the
western side of the lake near the Dapugang River mouth. Environmental variables (temperature,
dissolved oxygen, pH, total dissolved solids (TDS), and chlorophyll a) were measured in situ at
each site using a YSI 6600 multi-sensor sonde.
Water samples were collected in August 2013 (late summer bloom), June 2014 (early
summer bloom), March 2015 (no bloom/early spring bloom), and July 2016 (mid-summer
bloom). This sampling pattern was chosen to investigate seasonal changes of nutrients and
cyanobacteria in the lake. Water was collected into 4 l carboys at the surface (top 20 cm) and
near-bottom (approximately 2 m depth) to investigate any changes in nutrient dynamics
associated with depth. Samples for nutrient analyses ($NO_3^-$, $NO_2^-$, o-$PO_4^{3-}$, and urea) were
filtered immediately in the field using 0.2 µm nylon syringe filters (GE Millipore) into 15 ml
snap-cap tubes (Falcon) and stored frozen at -20°C. Nutrient samples were analyzed on a Lachat
QuikChem 8000 nutrient analyzer at the University of Texas Marine Science Institute (UTMSI;
Aug 2013, June 2014) or a Lachat 8500 nutrient analyzer at Wright State University (WSU;
March 2015, July 2016) according to manufacturer directions. Ambient $NH_4^+$ concentrations
were determined by ammonium retention time shift (AIRTS)  high performance liquid
chromatography (HPLC) at UTMSI (Gardner et al., 1995). Briefly, the atom % $^{15}N$ and total
$NH_4^+$ concentration are determined by comparing the retention time shift of the sample relative
to the natural abundance $NH_4^+$ standard (Gardner et al., 1996)
**2.2 Water column $NH_4^+$ uptake and regeneration**





NH$_4^+$ uptake and regeneration rates were determined following the protocol of McCarthy
et al. (2013). Water collected in 4 l carboys was returned to the Taihu Laboratory for Lake
Ecosystem Research (TLLER) for isotope amendments and incubations. 500 ml from each
site/depth was amended with 99.8% $^{15}$NH$_4$Cl (Isotec; concentration added 8–96 µM) and
distributed into six (triplicates for light and dark) 70 ml, clear tissue culture bottles (Corning;
McCarthy et al., 2007). Dark bottles were wrapped with thick aluminum foil. Initial samples (T$_0$)
were withdrawn from each bottle with a rinsed syringe, filtered (0.2 µm filters) immediately into
8 ml glass vials (Wheaton), and frozen until analysis at UTMSI. Light and dark bottles were then
submerged (approximate depth 0.2 m) in a mesh bag at in situ light and temperature in the lake.
After ~24 h, final samples (T$_f$) were filtered in the same manner as the T$_0$ samples. Total NH$_4^+$
concentrations and atom % $^{15}$N for all samples were determined by AIRTS/HPLC (Bruesewitz et
al., 2015; Gardner et al., 1995). Potential uptake and actual regeneration rates were calculated
using the Blackburn/Caperon isotope dilution model (Blackburn, 1979; Caperon et al., 1979;
McCarthy et al., 2013). The uptake rate is considered a potential rate, which includes
nitrification, assimilation, and other consumption processes, and regeneration encompasses
remineralization, decomposition of dead organic matter, heterotrophic excretion, respiration,
biodegradation, and sloppy feeding by zooplankton (Saba et al., 2011).
**2.3 Ammonia and nitrite oxidation rates**
Ammonia and NO$_2^-$ oxidation rates were measured directly using the $^{15}$NH$_4^+$ tracer
addition method. 500 ml of water from each station and depth was distributed into 750 ml
polycarbonate bottles, enriched with a tracer amount (approximately 20% of the total pool) of
99.8% $^{15}$NH$_4$Cl (Isotec), mixed thoroughly by inverting 10 times, and distributed into three 125
ml polycarbonate incubation bottles. Unenriched samples for each station and depth were



distributed into 125 ml incubation bottles. Initial samples ($T_0$) were filtered using 0.22 µm
syringe filters into 30 ml polycarbonate bottles and frozen until analysis. Final samples were
collected as described after incubating for 24 h at in situ light and temperature. Samples were
returned frozen to WSU for analysis.

Ammonia oxidation rates were measured from accumulation of $^{15}NO_2^-$ using the sodium

azide ($NaN_3$) reduction method (Heiss and Fulweiler, 2016; McIlvin and Altabet, 2005; Newell
et al., 2011). Briefly, 7.5 ml from each sample was distributed into a 12 ml Exetainer vial
(Labco, UK) and capped tightly. Each sample was then injected (with gastight syringe) with 0.25
ml of 1:1 (v:v) 2 M $NaN_3$ :20% $CH_3COOH$ solution (previously purged with Ar for 30 min),
followed by incubation for 1 h at 30 °C (McIlvin and Altabet, 2005). All $NO_2^-$ accumulated in
the sample from $NH_3$ oxidation was transformed chemically to $N_2O$. After 1 h, the reaction was
stopped by injection of 0.15 ml of 10 M NaOH.

Nitrite oxidation rates were measured from accumulation of $^{15}NO_3^-$ using the Cd

reduction/$NaN_3$ reduction method (Heiss and Fulweiler, 2016). Approximately 25 ml from each
sample was transferred into 50 ml centrifuge tubes. First, in situ $NO_2^-$ was removed with 0.25 ml
of 0.4 M sulfamic acid ($H_3NSO_3$). After 10 min, the reaction was neutralized with 0.125 ml of 2
M NaOH (Granger and Sigman, 2009). $NO_3^-$ was reduced to $NO_2^-$ by addition of 100 mg of
MgO, 6.6 g of NaCl, and 0.75–1 g of acidified Cd powder to each sample, followed by 17 h
incubation on a shaker table (McIlvin and Altabet, 2005). Samples were centrifuged at 2000 rpm
for 15 min, and 7.5 ml of supernatant was carefully transferred into 12 ml Exetainers. Cadmium-
reduced $NO_2^-$ was further reduced to $N_2O$ with the previously described $NaN_3$ method.

Samples for $NH_3$ and $NO_2^-$ oxidation were sent inverted to the University of California

Davis Stable Isotope Facility for isotopic analysis of $^{45/44}N_2O$ using a ThermoFinnigan GasBench





+ PreCon trace gas concentration system interfaced to a ThermoScientific Delta V Plus isotope-
ratio mass spectrometer (Bremen, Germany). Ammonia and $NO_2^-$ oxidation rates were corrected
for $NaN_3$ reduction efficiency, and $NH_3$ oxidation was calculated as:

$NH_3$ Ox (in nM day$^{-1}$) = $((^{15}N/^{14}N * [NO_2^-])_{24h} - (^{15}N/^{14}N * [NO_2^-])_{0h})/ \alpha * t$

Where $\alpha = [^{15}NH_4^+] / ([^{15}NH_4^+] + [^{14}NH_4^+])$
For $NO_2^-$ oxidation:

$NO_2^-$ Ox (in nM day$^{-1}$) = $((^{15}N/^{14}N * [NO_3^-])_{24h} - (^{15}N/^{14}N * [NO_3^-])_{0h})/ \alpha * t$

Where $\alpha = [^{15}NO_2^-] / ([^{15}NO_2^-] + [^{14}NO_2^-])$
**2.4 Quantitative Polymerase Chain Reaction (qPCR)**

During the 2014–2016 sampling events, environmental DNA for AOO abundance was

collected using 0.2 µm Sterivex filters (EMD Millipore, MA, USA) and preserved with Ambion
RNAlater (Invitrogen, Carlsbad, CA, USA). Approximately 60–120 ml of site water was pushed
through the filter for each station and depth and then stored filled with 5 mL RNAlater.
Preserved filters were frozen at -80 ˚C and transported to WSU. DNA was extracted using the
Gentra PureGene kit (Qiagen Inc., USA) extraction protocol with slight modifications (Newell et
al., 2011). Sterivex filters were first washed with Phosphate Buffer Saline 1X Solution (Fisher
BioReagents, USA) to remove any residual RNAlater. Lysis buffer (0.9 ml) and Proteinase K (4
µl) were added to the filters, followed by 1 h incubation at 55 °C and 1 h incubation at 65 °C.
The solution was removed to a 1.5 ml tube, and the incubation was repeated with fresh lysis
buffer and Proteinase K.

Concentration and purity of the DNA were measured spectrophotometrically (Nanodrop

2000, ThermoScientific). AOA were targeted with Arch-amoAF and Arch-amoAR primers
targeting the 635 base pair (bp) region of the *amoA* gene, subunit A of the ammonia



monooxygenase enzyme (AMO; Francis et al. 2005). Bacterial *amoA* was quantified using
amoAF and amoA2R primers (Rotthauwe et al., 1997) to target the 491 bp region of *amoA*.
qPCR standards were prepared by cloning the fragment of interest for AOA and AOB with the
TOPO TA Cloning Kit (Invitrogen, USA), inserting it into a competent cell plasmid (One Shot
E. coli cells, Invitrogen, USA), and isolating the plasmid containing the *amoA* gene using the
UltraClean Standard Mini Plasmid Prep Kit (Mo Bio Laboratories Inc., Carlsbad, CA, USA).

AOA and AOB qPCR assays were conducted within a single 96 well plate for each year

(2014, 2015, and 2016). Each run included three negative controls (no template), five standards
from serial dilution in triplicates, and the environmental DNA samples in triplicate. Each sample
and standard received 12.5 µl of SYBR green Fast Mastermix (Qiagen Inc., USA), 0.5 µl of each
100 µM primer, and 2–15 ng of template DNA.

All PCR work was performed in a PCR fume hood after cleaning the surface with

DNAaway (ThermoScientific, USA) and engaging the UV light (20 min) to prevent
contamination. qPCR protocol followed the method of Bollmann et al. (2014) for AOA (95 ℃
initial denaturation for 5 min, 95 ℃ denaturation for 30 sec, 53 ℃ annealing for 45 sec, and 72
℃ extension for 1 min; 45 cycles) and AOB (95 ℃ initial denaturation for 5 min, 95 ℃
denaturation for 30 sec, 56 ℃ annealing for 45 sec, 72 ℃ extension for 1 min; 45 cycles),
followed by the melting curve. Automatic settings for the thermocycler (Realplex, Eppendorf)
were used to determine threshold cycle (Ct values), efficiency (85–95%), and a standard curve
with $R^2$ values above 0.9. Gene copy number was calculated as (ng * number mol$^{-1}$)/ (bp * ng g$^{-1}$
$^{1}$ * g mol$^{-1}$ of bp)
and is reported in gene copies/ml of sample water.
**2.5 Statistical analysis**





All statistical analyses were performed using RStudio software (R Version 3.3.1). Prior to
statistical analysis, data were checked for normality using the Shapiro–Wilk normality test. The
only variables that were normally distributed were DO, pH, and TDS. To explore potential
environmental drivers of the rates, a multivariate correlation analysis was performed using the
Kendall correlation method for nonparametric data. A p-value of <0.05 was considered
statistically significant. Additionally, stepwise multiple regression models were run using the
MASS package (R Version 7.3). The best fitting model was selected based on the minimum
Akaike's Information Criteria (AIC; Akaike 1974). To normalize data for parametric analysis, all
non-normally distributed variables were $\log(x+1)$ transformed prior running the model.
**3.  Results**
**3.1 Lake ambient conditions**
Physicochemical parameters in Taihu varied seasonally (Table 1). The most pronounced
seasonal variations were observed in temperature and DO, with highest water temperature
recorded in August. DO varied significantly, with highest values in March and lowest in August
$(p < 0.01)$. pH varied significantly with season, with lowest values in March and highest in
August $(p < 0.01)$. TDS values were highest in July 2016 and lowest in August 2013 $(p < 0.001)$.
Ammonium concentrations remained high throughout all sampling events, with highest
values in March 2015 and lowest values in August 2013, but differences were not statistically
significant $(p = 0.125)$. Nitrite concentrations were not different between seasons, although they
were significantly higher at Station 10 than other stations $(p < 0.001)$. Nitrate concentrations
followed the pattern of $NH_4^+$ concentrations and were highest in March 2015 and lowest in
August 2013 $(p < 0.001)$. Orthophosphate concentrations followed a seasonal pattern with lowest





concentrations in March and highest in August ($p < 0.005$), and o-$PO_4^{3-}$ concentrations at Station
10 were significantly higher than at any other station ($p < 0.001$).
**3.2 Potential $NH_4^+$ uptake**
In August 2013, light uptake rates (all $NH_4^+$ uptake are potential rates) were uniform
across sites (mean = $0.40 \pm 0.04$ µmol $L^{-1}$ $hr^{-1}$) and did not vary between surface and bottom
waters (Fig. 2a). During the early summer bloom in June 2014, light uptake rates in surface
waters at Stations 1, 7, and 10 (mean = $0.80 \pm 0.06$ µmol $L^{-1}$ $hr^{-1}$) were significantly higher than
deep rates (mean = $0.31 \pm 0.08$ µM $hr^{-1}$; $p < 0.001$). However, light uptake rates at Station 3 did
not differ from zero at either depth (Fig. 2a). Mean surface and deep uptake rates in the dark in
August 2013 ($0.25 \pm 0.01$ µmol $L^{-1}$ $hr^{-1}$) and June 2014 ($0.13 \pm 0.05$ µmol $L^{-1}$ $hr^{-1}$) were
significantly lower than light uptake rates (Fig. 2b; $p < 0.05$). In the early spring bloom of March
2015, light uptake rates at Stations 1–7 (mean = $0.12 \pm 0.04$ µmol $L^{-1}$ $hr^{-1}$)were lower than those
during the August and June summer bloom (mean = $0.43 \pm 0.41$ µmol $L^{-1}$ $hr^{-1}$) except for Station
10, where the rates were significantly higher (mean = $1.36 \pm 0.20$ µmol $L^{-1}$ $hr^{-1}$; $p < 0.001$). In
contrast to summer, dark uptake rates in March 2015 were not significantly different than light
rates (Fig. 2b). During the July 2016 bloom, light uptake rates were highest at Stations 1, 7, and
10 ($1.31 - 6.82$ µmol $L^{-1}$ $hr^{-1}$). Stations 3 and 7 rates were highest in bottom waters ($0.80 \pm 0.16$
µmol $L^{-1}$ $hr^{-1}$ and $2.55 \pm 0.14$ µmol $L^{-1}$ $hr^{-1}$, respectively). In July 2016, light and dark uptake
rates did not differ significantly ($p = 0.15$); highest dark uptake rates were observed at Station 1
in surface water ($3.33 \pm 0.67$ µmol $L^{-1}$ $hr^{-1}$). Light uptake rates, across all stations and seasons,
correlated positively with TDS and $NH_4^+$:$NO_3^-$ and negatively with pH, while dark uptake rates
correlated positively with TDS, $NH_4^+$, and $NH_4^+$:$NO_3^-$, and negatively with pH (Table 2).
**3.3 Regeneration of $NH_4^+$**





Regeneration rates in the light and dark (all $NH_4^+$ regeneration rates are actual rates, not

potential) were not significantly different from each other across all years and seasons; therefore,
light and dark rates were averaged together (Fig. 2c). Regeneration rates did not differ
significantly between the summer bloom sampling events in August 2013 and June 2014 (mean
$= 0.22 \pm 0.03$ µmol $L^{-1}$ $hr^{-1}$), but July 2016 regeneration rates (mean = $0.75 \pm 0.16$ µmol $L^{-1}$ $hr^{-1}$)
were significantly higher than in August and June ($p = 0.004$), with exceptionally high
regeneration rates occurring in surface waters in July at Station 1 (mean = $2.37 \pm 0.16$ µmol $L^{-1}$
$hr^{-1}$). In March 2015, mean surface and deep regeneration rates decreased from the river mouth
(Station 10; $0.88 \pm 0.15$ µmol $L^{-1}$ $hr^{-1}$) towards the center of the lake, with significantly higher
regeneration rates at 10 than Stations 1–7 (mean = $0.10 \pm 0.03$ µmol $L^{-1}$ $hr^{-1}$; $p < 0.01$).
Regeneration rates were positively correlated with TDS, $NH_4^+$, and o-$PO_4^{3-}$ concentrations, and
$NH_4^+$:$NO_3^-$ (Table 2).
**3.4 Nitrification (2014-2016)**

Note that nitrification rates are presented in nmol $L^{-1}$ $d^{-1}$ for consistency with literature

reported values. At Stations 1, 3, and 7, $NO_2^-$ oxidation rates (mean = $248 \pm 39.0$ nmol $L^{-1}$ $d^{-1}$)
exceeded $NH_4^+$ oxidation rates (mean = $21.9 \pm 6.34$ nmol $L^{-1}$ $d^{-1}$; $p < 0.001$) by an order of
magnitude across all sampling events (Fig. 3a). The total nitrification rates at Station 3 did not
vary across seasons. At Station 7 in the central lake, highest total nitrification rates were
observed in March (mean = $663 \pm 69.4$ nmol $L^{-1}$ $d^{-1}$) in both surface and deep waters compared
to the lowest rates in July 2016 (mean = $1.58 \pm 0.78$ nmol $L^{-1}$ $d^{-1}$). At Station 1, the highest rates
were measured in surface waters in July (mean = $773 \pm 50.7$ nmol $L^{-1}$ $d^{-1}$), but the rates at depth
followed a seasonal pattern from high in the spring (mean = $646 \pm 158$ nmol $L^{-1}$ $d^{-1}$) to an order
of magnitude lower in the summer (mean = $9.86 \pm 3.28$ nmol $L^{-1}$ $d^{-1}$).



Total nitrification rates at Station 10 were significantly higher than other stations (Fig.
3b; p < 0.001). Rates were, at times, orders of magnitude higher, and total nitrification ranged
from 148 – 3750 nmol L$^{-1}$ d$^{-1}$ (mean = 1590 ± 1390 nmol L$^{-1}$ d$^{-1}$), compared to Stations 1–7
ranging from 2.00 – 771 nmol L$^{-1}$ d$^{-1}$ (mean = 270 ± 277 nmol L$^{-1}$ d$^{-1}$). While NO$_2^-$ oxidation
rates exceeded NH$_4^+$ oxidation rates in 2014 and 2015, NH$_4^+$ oxidation rates in July 2016 were
significantly higher than other years (1650 ± 55.0 nmol L$^{-1}$ d$^{-1}$; p < 0.001).
**3.5 Ammonia oxidizer abundance**
Abundance of the bacterial *amoA* gene for all years (2014–2016) varied between
undetectable to 2.85 x 10$^5$ ± 5.20 x 10$^4$ copies ml$^{-1}$. Archaeal *amoA* abundance ranged from
undetectable to 1.03 x 10$^7$ ± 3.37 x 10$^6$ copies ml$^{-1}$ (Fig. 4a). Neither AOB nor AOA *amoA* gene
copy abundances were statistically different between the three seasons. The highest ratio of
AOB:AOA gene abundance was reported at Station 3 in Meiliang Bay (1.81; Fig. 4b) and lowest
in the open lake (0.01; Station 7). AOB gene abundance was positively correlated with NH$_4^+$,
NO$_2^-$, and o-PO$_4^{3-}$ concentrations, and NH$_4^+$:NO$_3^-$, while AOA gene abundance was not
significantly correlated to any environmental variable (Table 2).
**4.   Discussion**
**4.1 Ammonium regeneration and potential uptake**
Ammonium uptake rates (0.02 – 6.82 µmol L$^{-1}$ hr$^{-1}$) reported here were within the range of or
slightly higher than rates reported in other studies (Table 3). Rates were higher than uptake rates
reported previously in Meiliang Bay (0.11 – 1.54 µmol L$^{-1}$ hr$^{-1}$) and the central lake (0.03 – 0.32
µmol L$^{-1}$ hr$^{-1}$) but within the range of rates reported in the Liangxihe River (0.70 – 4.19 µmol L$^{-1}$
hr$^{-1}$; McCarthy et al., 2007). Light uptake rates in March, June, and August resembled rates in
eutrophic Lake Okeechobee but were higher than rates in Missisquoi Bay, Lake Champlain,



Lake Michigan, eutrophic New Zealand lakes Rotorua and Rotoiti, and the Mississippi River
plume (Table 3 and references therein). Higher light uptake rates were reported in
hypereutrophic Lake Maracaibo, Venezuela, and a Lake Erie coastal wetland, Old Woman Creek
(Table 3). Light uptake rates were marginally higher ($p = 0.08$) than dark uptake rates,
presumably due to photosynthetic phytoplankton activity. Photoautotrophic uptake is greatly
reduced in the dark, so dark uptake rates can be attributed mostly to heterotrophic or
chemolithoautotrophic organisms. However, photoautotrophs may assimilate nutrients in the
dark under nutrient limitation (Cochlan et al., 1991). Uptake rates were significantly higher in
July 2016 than at other times, which may have been due to higher precipitation and subsequent
runoff; during summer 2016, average rainfall in June and July was about 305 mm compared to
106 mm in June 2014, 105 mm in August 2013, and 54 mm in March 2015
(WorldWeatherOnline.com; accessed on <08/02/2017>). Dark uptake rates in Taihu exceeded
dark rates reported in Lake Okeechobee ($0.02 - 0.04$ µmol $L^{-1}$ $hr^{-1}$; James et al. 2011),
Missisquoi Bay, Lake Champlain ($0.10$ µmol $L^{-1}$ $hr^{-1}$; McCarthy et al., 2013), and Lake
Michigan (7 nmol $L^{-1}$ $hr^{-1}$; Gardner et al., 2004) suggesting increased activity of both
heterotrophs and chemolithoautotrophs in Taihu. A previous metagenomics study of the bloom
composition in Taihu revealed an overlooked contribution of heterotrophic bacteria to N
assimilation processes by *Microcystis*, which could be important in driving toxic blooms (Steffen
et al., 2012).

Internal $NH_4^+$ cycling via regeneration is important in Taihu and varies seasonally (McCarthy

et al., 2007; Paerl et al., 2011). In March 2015, about 38% of light uptake for all sites and depths
was supported by regeneration (Fig. 2d). This proportion increased in June 2014 and July 2016
to 58% and 42%, respectively, and was highest in August 2013, when regeneration could



account for 109% of uptake. The importance of regeneration corresponded to decreasing in situ
$NH_4^+$ concentrations (Fig. 2D). These results suggest that, in spring and early summer,
regeneration supplemented the ambient $NH_4^+$ in the water column to support algal production,
whereas in the late summer, cyanoHABs relied heavily on $NH_4^+$ from regeneration to sustain
blooms. Water column regeneration may supply more $NH_4^+$ for blooms than sediment $NH_4^+$
regeneration in Taihu due to combined spatial, temperature, and biogeochemical factors
(McCarthy et al., 2007; Gardner et al., 2017). Rapid decomposition of cyanoHABs biomass may
provide $NH_4^+$ for nitrification, which provides substrate for denitrification. High rates of
sediment denitrification (McCarthy et al., 2007) may lead to increased N limitation at the end of
the bloom season in late summer and fall (Paerl et al., 2011; Xu et al., 2010)

To calculate whole-lake, water column $NH_4^+$ regeneration rates, we divided the lake (2,338

$km^2$; Qin et al., 2007) into four different sections based on geochemical and ecological properties
(Qin, 2008): (1) three northern bays (361.8 $km^2$; depth = 1.9 m) most affected by the blooms; (2)
the main lake (1,523.9 $km^2$; depth = 1.9 m); (3) the East Taihu region, dominated by rooted and
floating macrophytes (357.5 $km^2$; depth = 1.4 m); and (4) shorelines <1 m deep (94.8 $km^2$). We
considered regeneration rates from Stations 1 and 3 to represent the northern bays area, Station 7
as the main lake, Station 10 as shoreline, and regeneration rates previously reported for East
Taihu (McCarthy et al., 2007; Paerl et al., 2011). When extrapolated to the volume of these four
zones in Taihu, regeneration returned about 3.04 x $10^7$ kg of $NH_4^+$ annually in the three northern
bays, 6.71 x $10^7$ kg of $NH_4^+$ in the main lake, 8.87 x $10^6$ kg of $NH_4^+$ along the shorelines, and
2.88 x $10^6$ kg of $NH_4^+$ in East Taihu Lake. These values sum to 1.09 x $10^8$ kg of $NH_4^+$ recycled
in the water column, approximately two times higher than reported external N loadings, which
range from 5.11 x $10^7$ and 7.00 x $10^7$ kg annually (Chen et al., 2012; Yan et al., 2011). This





rough estimate of lake-wide regeneration is based on rates measured at specific stations at
discreet times; improved spatial and temporal resolution of measurements are needed to improve
these estimates. Additionally, these calculated values are an overestimate given that most of the
rates measured and reported in this study are during spring and summer months, not fall and
winter. Taihu is a complex ecosystem with 172 rivers and channels connected to the lake (Qin et
al., 2007), making any estimations of total N loadings challenging. As such, we believe that the
reported total N loads to Taihu are likely an underestimate. However, our results show that these
external N loads are fueling high regeneration rates and suggest that microbial denitrification
cannot keep pace with external N loads. Increasing nutrient loadings can result in decreasing
efficiency of denitrification (Gardner and McCarthy, 2009; Mulholland et al., 2008), which will
limit the ability of a system to self-mitigate excess N loads.
**4.2 Nitrification**
Nitrification rates reported in this study exceeded previously reported rates in most
oligotrophic and mesotrophic freshwater systems. Published nitrification rates in lakes include
the water columns of saline Lake Mono, CA (60–480 nmol $L^{-1}$ $d^{-1}$; Carini and Joye, 2008), Lake
Okeechobee, FL (67–97 nmol $L^{-1}$ $hr^{-1}$; James et al., 2011), Lake Superior, USA (0–51 nmol $L^{-1}$
$d^{-1}$; Small et al., 2013), and in sediments of Lake Onondaga (0.37 g N $m^{-2}$ $d^{-1}$; Pauer and Auer,
2000). Similarly, only a few studies in freshwater systems report rates of ammonia oxidation
(Lake Superior (18–34 nmol N $L^{-1}$ $d^{-1}$); Small et al., 2013). Rates on this scale were previously
reported only in eutrophic Lake Mendota (WI; 1700 – 26000 nmol $L^{-1}$ $hr^{-1}$; Hall, 1986) and the
Paerl River Estuary (China; 2100 – 65100 µmol $L^{-1}$ $d^{-1}$; Dai et al., 2008). High nitrification rates
in Taihu can be attributed to high ambient $NH_4^+$ concentrations, up to 40 µM at Station 1 in 2016
and 135 µM at Station 10 in 2014. These high concentrations of $NH_4^+$ are due to high external N



loadings, including N in organic matter, into the lake, of which ~$1.32 \times 10^7$ kg were loaded as
$NH_4^+$ in 2009 (Yan et al., 2011). The significant relationships between nitrification and $NH_4^+$
oxidation with $NH_4^+$, $NO_2^-$, and $NO_3^-$ concentrations and between $NH_4^+$ oxidation and
$NH_4^+$:$NO_3^-$ (Table 2) support these observations.
Ammonia oxidation rates were positively correlated with ambient $NH_4^+$, $NO_2^-$, and $NO_3^-$
concentrations ($p < 0.005$; Table 2), as expected. Substrate concentrations drive $NH_4^+$ oxidation
rates and therefore end-product pools, since it is the rate limiting step of nitrification (i.e.,
completion of nitrification is dependent on the first step). Nitrite oxidation rates, however, were
an order of magnitude higher than $NH_4^+$ oxidation rates and were correlated with ambient $NH_4^+$
and $NO_3^-$ concentrations. Higher $NO_2^-$ oxidation rates were expected, since $NO_3^-$ is the product
of $NO_2^-$ oxidation, and $NO_2^-$ oxidation relies on the product of $NH_4^+$ oxidation. Nitrite oxidation
rates were not related to $NO_2^-$ concentrations, perhaps due to the standing pool of ambient $NO_2^-$.
However, at some stations, ambient $NO_2^-$ was substantial (e.g., Station 10, June 2014; 14 – 15
µM). This accumulation of $NO_2^-$ could indicate that $NO_2^-$ oxidizers were saturated, as reported
$K_m$ values for $NO_2^-$ oxidation in an oligotrophic, oxygen deficient region in the ocean were 0.25
$\pm$ 0.16 µM (Sun et al., 2017). However, culture experiments report $K_m$ values ranging from 6–
544 µM for *Nitrospira, Nitrobacter,* and *Nitrotoga spp.* (Blackburne et al., 2007; Nowka et al.,
2015; Ushiki et al., 2017).
At most stations, nitrification rates in Taihu were highest in March, lower in June, and lowest
in July. During the spring sampling, nitrification accounted for about 8% of light uptake and
15% of dark uptake at Stations 1 – 7. In June, nitrification accounted for 2.6% of light uptake
and 9.6% of dark uptake, and in July only 0.2% and 0.3% of light and dark uptake, respectively.
These results show a seasonal trend of decreasing contribution of nitrification to total uptake





rates and higher contribution of nitrifiers to dark uptake. Chemolithoautotrophs (including
nitrifiers) do not rely on light for energy and continue to assimilate $NH_4^+$ in dark conditions.
Phytoplankton, including cyanobacteria, can also assimilate $NH_4^+$ in the dark, especially when
nutrients are limiting (Cochlan et al., 1991), and N has been shown to limit primary production
in Lake Taihu, especially in summer (e.g., Paerl et al., 2011). However, the presence of high
dissolved inorganic N concentrations in ambient water samples suggests that the observed dark
uptake was likely performed primarily by non-photoautotrophs, including nitrifiers.

We observed no significant seasonal change in nitrification across all stations and no

consistent pattern between temperature and nitrification. While the lack of relationship of
nitrification with temperature agrees with nitrification studies in the ocean (Ward, 2008), other
studies have reported temperature as a potential driver of nitrification in coastal waters (Heiss
and Fulweiler, 2016). While not statistically linked to changes in temperature, the contribution of
nitrification to total uptake rates decreased in summer, likely as a result of competition with the
*Microcystis* bloom and associated heterotrophic bacteria. Non-$N_2$ fixing cyanobacteria, including
*Microcystis,* are exceptional competitors for $NH_4^+$ (Blomqvist et al., 1994). With a high
saturation threshold and reported $K_m$ values of 26.5 µM (Nicklisch and Kohl 1983) and 37 µM
(Baldia et al., 2007), it can outcompete nitrifiers at the high ambient $NH_4^+$ concentrations in
Taihu. Additionally, *Microcystis* can regulate its buoyancy and scavenge nutrients throughout the
water column to effectively compete for light with other phytoplankton (Brookes and Ganf,

2001).

Nitrification at the river mouth Station (10) differed dramatically from other stations. Total

nitrification rates and $NH_4^+$ oxidation rates were, at times, orders of magnitude higher than at
other stations. Also, Station 10 did not follow the trend of decreasing nitrification contribution





with the bloom. Nitrification accounted for 19% of light uptake and 64.8% of dark uptake in
June and only 1.7% and 2%, respectively, in March. These very high $NH_4^+$ oxidation rates, along
with high ambient $NH_4^+$ and $NO_2^-$ concentrations, suggest that $NH_4^+$ and $NO_2^-$ oxidation could be
uncoupled at this station. We speculate that Station 10 differs from other stations because of the
large nutrient and suspended particle loads from the Dapugang River, the second largest inflow
into the lake (Yan et al., 2011). Suspended particles from sediments could trigger heterotrophic
and anaerobic processes at Station 10, including reduction of $NO_3^-$ to $NO_2^-$ (Krausfeldt et al.,
2017; Yao et al. 2016). In fact, denitrification and anammox gene transcripts were observed
recently in the water column at Station 10 (Krausfeldt et al., 2017). These authors also speculated
that the discharge of suspended sediments from the river might play a role in coupling anaerobic
and aerobic processes in the turbid water column, resulting in rapid cycling of reduced and
oxidized forms of N. Nitrification is the link between introduction of reduced N into the system
and the removal of N through denitrification. Therefore, the efficiency of nitrification is crucial
to the removal of N from this hypereutrophic lake.
**4.3 Ammonia oxidizer abundance**
AOB and AOA coexist in the environment, and environmental variables shape the
community structure. AOA often dominate in environments with low substrate concentrations,
such as the open ocean or oligotrophic lakes (Beman et al., 2008; Bollmann et al., 2014; Newell
et al., 2011), while AOB are often more abundant in nutrient rich waters and soils (Hou et al.,
2013; Jia and Conrad, 2009; Kowalchuk and Stephen, 2001; Verhamme et al., 2011). This
substrate concentration adaptation is dictated by different physiological abilities to assimilate
$NH_4^+$. Culture studies show that AOA have a very high affinity (low half saturation constant;
$K_m$) for $NH_4^+$, and in general are saturated faster than AOB (Martens-Habbena et al., 2009). The



low half saturation constant ($K_m$ = 0.132 µM; Martens-Habbena et al., 2009) of AOA gives them
a competitive advantage in low $NH_4^+$ conditions. In contrast, the high $K_m$ of AOB (10–1000 µM)
allows them to assimilate more $NH_4^+$ before becoming fully saturated, an advantage for higher
$NH_4^+$ concentration conditions. Although oligotrophic AOA appear to proliferate in the
environment (Francis et al., 2005), some species adapt to higher substrate concentrations (Jung et
al., 2011; Tourna et al., 2011).
Results from the *amoA* gene copy abundance analysis show that AOA were more abundant
than AOB across all stations and seasons in Taihu. Although this result does not support our
original hypothesis, the results agree with previous studies in the water column and sediments in
Taihu (Zeng et al., 2012), which reported higher AOA abundance (4.91 x $10^5$ – 8.65 x $10^6$ copies
$g^{-1}$ sediment) than AOB (3.74 x $10^4$ – 3.86 x $10^5$ copies $g^{-1}$ sediment) in Meiliang Bay. Similarly,
another Taihu sediment study showed more AOA than AOB in sediments at all 20 investigated
stations (Wu et al., 2010).
The differences in abundance of AOO between stations, represented as AOB:AOA, show
spatial variability between the more nearshore and central lake stations (Fig. 4b). In this study,
AOA were more abundant in the central lake (Station 7), whereas AOB were more abundant
closer to shore. Due to a higher affinity for substrate (lower $K_m$), AOA are likely more
competitive when nutrient concentrations are lower, such as in the open lake (mean offshore
$NH_4^+$ concentration = 3.69 µM). In contrast, AOB, with higher $K_m$, thrive at higher $NH_4^+$
concentrations at nearshore locations (mean nearshore $NH_4^+$ concentration = 31.3 µM). These
results agree with previous research in Taihu, where AOA outnumbered AOB in sediments at
mesotrophic sites, and AOB were more abundant at hypereutrophic locations (Hou et al., 2013).
Another study in Taihu sediments also reported that both AOA abundance and AOA:AOB were





negatively correlated with ambient $NH_4^+$ concentration (Wu et al., 2010). However, the data
reported in this study show no significant relationship between AOA and $NH_4^+$, $NO_2^-$, and $NO_3^-$
(Table 2).

Despite AOA outnumbering AOB, AOB abundance was correlated with total nitrification

rates for all stations and all seasons ($p < 0.005$), but AOA abundance was not. This result agrees
with a previous study in Taihu sediments, where AOA were negatively correlated ($r = 0.53$, $p <$
$0.05$) with potential nitrification rates ($0 - 3.0 \mu g$ $NO_3$—$N$ $g^{-1}$ dry sediment; Hou et al., 2013). We
speculate that AOA oxidized $NH_4^+$ at lower rates due to oversaturation and inhibition and may
not have contributed as much as AOB to total nitrification rates in our study. This conclusion
was also reached in Plum Island Sound (MA, USA), where abundance of archaeal *amoA* was
higher than bacterial, but potential nitrification rates did not correlate with AOA (Bernhard et al.,
2010). The authors hypothesized various scenarios, including inhibition of AOA due to high
substrate concentrations, competition for $NH_4^+$ with AOB, or AOA using an alternative energy
source (Bernhard et al., 2010). Our results support the interpretation that AOA are at a
disadvantage when competing with AOB for $NH_4^+$ in a hypereutrophic system and most likely
did not play a major role in observed nitrification in Taihu. Recently studies show that AOA can
oxidize cyanate (Palatinszky et al., 2015) and urea (Tolar et al., 2016). Therefore, we speculate
that AOA might be playing a different role in Taihu.
**4.4 Multiple regression model**

The best-fitting multiple regression models for N dynamics in Taihu (Table 4) supported

the Kendall non-parametric analysis (Table 2). Ammonium uptake and regeneration rates and
nitrification were driven by ambient $NH_4^+$, $NO_2^-$, and $NO_3^-$ concentrations. Additionally, the
best-fitting models revealed that variables that changed with season had major influences on the




520 models (Table 4). For example, uptake in the light and dark and regeneration rates were

521 positively influenced by temperature and DO and negatively by pH. However, models for

522 nitrification rates, and $NH_4^+$ and $NO_2^-$ oxidation rates, did not reveal that the seasonal variables,

523 such as temperature, played a major role in the model.

## 5. Conclusions

525 This study highlights the importance of water column $NH_4^+$ regeneration and nitrification

526 in bloom formation and maintenance in Taihu. We showed that nitrification rates were detectable

527 during the bloom but decreased as the bloom progressed, suggesting that nitrifiers are weaker

528 competitors for substrate than *Microcystis*. Also, seasonal changes in light and dark $NH_4^+$ uptake

529 and nitrification rates showed that AOO are outcompeted by *Microcystis*. Extremely high

530 nitrification rates at the river mouth (Station 10) differed from rates at other stations, suggesting

531 that other processes, such as coupled nitrification/denitrification, might be important in

532 suspended sediments. Previous studies reported coupled denitrification with nitrification in

533 sediments (McCarthy et al., 2007). Functional gene analysis suggested that gene abundance does

534 not necessarily reflect performance of the function in eutrophic lakes. We speculate that AOA

535 are present in the lake but do not contribute proportionately to nitrification, suggesting that AOA

536 might play another role in the lake.

537 Ammonium inflow into the lake is a large source of reduced N, but external inputs are

538 not the sole source. Extrapolated whole-lake regeneration rates in the water column were twice

539 as high as external N loadings into the lake. To mitigate harmful algal blooms, N loadings into

540 the lake must be reduced so that N can be efficiently removed through denitrification, instead of

541 being recycled in the water column. Our results support the recent calls for dual nutrient (N + P)



management strategies (Paerl et al., 2011) and highlight the importance of (chemically) reduced
N removal through nitrification and denitrification.
Acknowledgments
We thank Guang Gao for laboratory space at NIGLAS and Kaijun Lu and other graduate
students at NIGLAS and TLLER for help in the field and in the lab. We also thank Richard
Doucett at UC Davis Stable Isotope Facility for $^{15}$N sample analysis, and Justin Myers, Megan
Reed, Ashlynn Boedecker, and Desi Niewinski at WSU for help with nutrient analysis. We also
thank Daniel Hoffman at WSU for valuable help with nitrification experiments and Elise Heiss
for her input on statistical analysis. This work was jointly supported by the International Science
& Technology Cooperation Program of China (2015DFG91980) and the National Natural
Science Foundation of China (41573076, 41771519).





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



Figure list

Figure 1. Map of sampling stations in Taihu (modified from Paerl et al. 2011).

Figure 2. Ammonium dynamics in Taihu. (a) potential light uptake rates ± one standard error. (b)
potential dark uptake rates ± one standard error. (c) Mean light and dark regeneration rates ± one
standard error. (d) Seasonal averaged percent of light uptake supported by regeneration ± one
standard error and averaged in situ $NH_4^+$ concentrations.


Figure 3. Ammonia and nitrite oxidation rates (total nitrification) ± one standard deviation. (a)
Stations 1–7. (b) Station 10. Note the difference in y-axis scale.


Figure 4. Ammonia oxidizing organism population characteristics. (a) Ammonia oxidizer
abundance (DNA) ± one standard deviation. (b) Ratio of abundance of AOB to AOA.








Table 1.
Environmental characteristics during sampling events for each station/depth: temperature,
dissolved oxygen (DO), pH, chlorophyll a (chl a; surface only), total dissolved solids (TDS), and
in situ nutrient concentrations. S in station name = surface water (0.2 m), and D = deep, near-
bottom water (~2 m).

| Year/ Month | Station | Temp (°C) | DO (mg L$^{-1}$) | pH | Chl a (µg L$^{-1}$) | TDS | [NH$_4^+$] (µM) | [NO$_2^-$] (µM) | [NO$_3^-$] (µM) | [PO$_4^{3-}$] (µM) |
|---|---|---|---|---|---|---|---|---|---|---|
| 2013 | 1S | 30.9 | 3.53 | 8.11 | 53.99 | 377 | 1.37 | 0.28 | 2.09 | 2.51 |
| August | 1D | 30.8 | 4.24 | 8.05 | | 377 | 1.79 | 0.23 | 2.17 | 2.96 |
| | 3S | 32.5 | 9.07 | 9.02 | 57.62 | 390 | 0.51 | 0.23 | 1.84 | 1.64 |
| | 3D | 31.9 | 7.4 | 8.97 | | 390 | 0.56 | 0.25 | 0.60 | 1.62 |
| | 7S | 30.4 | 3.4 | 8.05 | 22.16 | 357 | 0.26 | 0.21 | 2.20 | 0.41 |
| | 7D | 30.4 | 3.4 | 8.18 | | 357 | 0.32 | 0.14 | 0.90 | 2.73 |
| | 10S | 32.1 | 8.6 | 9.33 | 40.82 | 375 | 0.61 | 1.90 | 7.74 | 4.83 |
| | 10D | 32.0 | 8.0 | 9.43 | | 375 | 0.29 | 1.04 | 3.76 | 5.69 |
| 2014 | 1S | 23.9 | 8.5 | 8.11 | 13.66 | 436 | 6.16 | 3.33 | 87.49 | 1.75 |
| June | 1D | 22.7 | 5.1 | 8.07 | | 437 | 8.34 | 3.36 | 87.09 | 0.69 |
| | 3S | 27.2 | 8.6 | 8.73 | 11.05 | 419 | 1.09 | 1.72 | 58.29 | 0.24 |
| | 3D | 25.4 | 7.3 | 8.71 | | 411 | 1.20 | 2.61 | 57.41 | 0.35 |
| | 7S | 22.8 | 9.7 | 7.85 | 42.41 | 383 | 1.55 | 0.83 | 66.32 | 0.39 |
| | 7D | 22.5 | 8.6 | 7.69 | | 384 | 1.59 | 0.74 | 61.59 | 2.13 |
| | 10S | 26.3 | 5.6 | 8.89 | 79.52 | 424 | 35.39 | 14.93 | 69.98 | 2.43 |
| | 10D | 26.4 | 5.5 | 8.6 | | 424 | 35.75 | 15.13 | 68.93 | 2.52 |
| 2015 | 1S | 11.6 | 10.05 | 8.34 | 7.48 | 393 | 2.49 | 0.55 | 53.89 | 0.20 |
| March | 1D | 11.7 | 3.4 | 6.67 | | 393 | 2.49 | 0.58 | 54.74 | 0.04 |
| | 3S | 9.4 | 12.8 | 7.74 | 20.37 | 414 | 0.00 | 0.82 | 119.44 | 0.03 |
| | 3D | 8.2 | 12.9 | 7.52 | | 414 | 0.83 | 0.86 | 117.61 | 0.05 |
| | 7S | 10.8 | 11.32 | 8.4 | 10.52 | 416 | 5.93 | 1.95 | 172.19 | 0.02 |
| | 7D | 10.7 | 10.75 | 8.01 | | 416 | 5.93 | 1.44 | 136.2 | 0.12 |
| | 10S | 9.6 | 8.87 | 7.94 | 6.00 | 422 | 131.48 | 7.05 | 270.59 | 1.41 |
| | 10D | 9.4 | 8.71 | 7.73 | | 421 | 131.84 | 6.97 | 269.47 | 1.36 |
| 2016 | 1S | 26.7 | 11.30 | 7.89 | 96.79 | 445 | 43.34 | 8.86 | 79.72 | 1.95 |
| July | 1D | 25.5 | 7.55 | 7.67 | | 458 | 20.03 | 6.71 | 58.78 | 1.31 |
| | 3S | 26.1 | 7.0 | 8.50 | 100.99 | 410 | 17.59 | 0.86 | 3.81 | 1.05 |
| | 3D | 26.3 | 7.3 | 8.50 | | 410 | 21.08 | 0.72 | 3.87 | 1.16 |
| | 7S | 25.8 | 10.03 | 7.95 | 13.22 | 465 | 0.33 | 0.08 | 16.39 | 0.03 |
| | 7D | 25.1 | 8.88 | 7.88 | | 466 | 0.25 | 0.11 | 16.52 | 0.05 |
| | 10S | 25.6 | 4.10 | 7.75 | 21.31 | 470 | 13.41 | 9.66 | 93.96 | 2.43 |
| | 10D | 23.4 | 4.10 | 7.62 | | 470 | 65.31 | 8.45 | 66.77 | 3.18 |




Table 2.
Details of non-parametric Kendall's correlation analysis. Statistically significant (p < 0.05) Kendall's Tau coefficients are bold.

| | | Temp | DO | pH | Chla | TDS | NH4 | NO2 | NO3 | P | NH4:NO3 |
|---|---|---|---|---|---|---|---|---|---|---|---|
| Uptake L | Kendall's T | -0.010 | -0.061 | **-0.326** | 0.133 | **0.321** | 0.230 | 0.020 | 0.048 | 0.081 | **0.301** |
| | p value | 0.935 | 0.626 | **0.009** | 0.471 | **0.010** | **0.064** | 0.871 | 0.697 | 0.517 | **0.016** |
| Uptake D | Kendall's T | -0.014 | -0.041 | **-0.293** | 0.117 | **0.337** | **0.295** | 0.000 | 0.069 | 0.069 | **0.369** |
| | p value | 0.910 | 0.745 | **0.019** | 0.529 | **0.007** | **0.018** | 1.000 | 0.581 | 0.581 | **0.003** |
| Regeneration | Kendall's T | 0.095 | -0.110 | -0.103 | 0.300 | **0.301** | **0.344** | 0.149 | 0.012 | **0.259** | **0.487** |
| | p value | 0.446 | 0.381 | 0.408 | 0.105 | **0.016** | **0.006** | 0.230 | 0.923 | **0.038** | **<0.001** |
| Nitrification | Kendall's T | -0.138 | -0.128 | -0.214 | 0.242 | -0.058 | **0.385** | **0.341** | **0.377** | **0.341** | 0.272 |
| | p value | 0.346 | 0.385 | 0.143 | 0.273 | 0.691 | **0.009** | **0.020** | **0.010** | **0.020** | 0.063 |
| $NH_4^+$ Ox. | Kendall's T | -0.080 | -0.200 | -0.011 | -0.030 | 0.139 | **0.575** | **0.514** | **0.406** | **0.543** | **0.461** |
| | p value | 0.585 | 0.172 | 0.941 | 0.891 | 0.345 | **<0.001** | **<0.001** | **0.005** | **<0.001** | **0.002** |
| $NO_2^-$ Ox. | Kendall's T | -0.106 | -0.081 | -0.197 | 0.333 | -0.077 | **0.325** | 0.266 | **0.281** | 0.266 | 0.277 |
| | p value | 0.471 | 0.585 | 0.180 | 0.131 | 0.602 | **0.027** | 0.070 | **0.056** | 0.070 | 0.059 |
| AOA | Kendall's T | 0.109 | 0.179 | 0.083 | 0.273 | 0.161 | 0.015 | -0.014 | -0.051 | 0.043 | -0.004 |
| | p value | 0.457 | 0.224 | 0.568 | 0.217 | 0.275 | 0.921 | 0.921 | 0.728 | 0.766 | 0.980 |
| AOB | Kendall's T | 0.175 | -0.157 | -0.149 | 0.273 | 0.175 | **0.458** | **0.341** | 0.130 | **0.500** | **0.425** |
| | p value | 0.234 | 0.286 | 0.309 | 0.217 | 0.233 | **0.002** | **0.020** | 0.372 | **0.001** | **0.004** |





Table 3.
Light (L) and dark (D) ammonium uptake and regeneration rates among different freshwater studies ± standard error.

|  | Uptake (L) | Uptake (D) | Reg Avg | Reference |
|---|---|---|---|---|
| Lake Lugano | 0.017 ± 0.001 | 0.008 ± 0.003 | 0.010 ± 0.002 | McCarthy unpublished |
| Lake Michigan | 0.019 ± 0.004 | 0.01 ± 0.002 | 0.008 ± 0.001 | Gardner et al. 2004 |
| Lake Rotorua | 0.114 ± 0.008 | 0.021 ± 0.005 | 0.047 ± 0.007 | McCarthy unpublished |
| Lake Rotoiti | 0.132 ± 0.033 | 0.08 ± 0.019 | 0.063 ± 0.018 | McCarthy unpublished |
| Missisquoi Bay | 0.205 ± 0.022 | 0.104 ± 0.015 | 0.085 ± 0.013 | McCarthy et al. 2013 |
| Lake Erie | 0.258 ± 0.128 | 0.036 ± 0.009 | 0.124 ± 0.052 | McCarthy unpublished |
| Lake Okeechobee | 0.577 ± 0.006 | 0.029 ± 0.01 | 0.160 ± 0.021 | James et al. 2011 |
| Taihu Lake | 0.655 ± 0.285 | 0.271 ± 0.111 | 0.325 ± 0.144 | McCarthy et al. 2007 |
| Taihu Lake | 0.886 ± 0.09 | 0.399 ± 0.121 | 0.368 ± 0.071 | This study |
| Old Woman Creek | 0.925 ± 0.223 | 0.223 ± 0.043 | 0.320 ± 0.074 | McCarthy et al. 2007 |

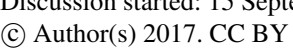



Table 4.
Details of best-fitting multiple regression models determined by stepwise regression. All rates, temperature, and ambient nutrient concentrations were log-transformed prior to analysis.

| Process | Variable | Parameter | | | Model | | |
|---------|----------|-----------|----|---|-------|---|---|
| | | Estimate | Std. estimate | P | Adj. $R^2$ | F | P |
| Uptake Light | T | 1.048 | 0.216 | 0.0001 | 0.643 | 10.3 | $9.14 \times 10^{-6}$ |
| | DO | 0.053 | 0.012 | 0.0002 | | | |
| | pH | -0.320 | 0.054 | 0.0000 | | | |
| | $NH_4^+$ | 0.669 | 0.272 | 0.0213 | | | |
| Uptake Dark | T | 0.488 | 0.121 | 0.0005 | 0.745 | 16.1 | $1.66 \times 10^{-7}$ |
| | DO | 0.034 | 0.007 | 0.0000 | | | |
| | pH | -0.187 | 0.031 | 0.0000 | | | |
| | $NH_4^+$ | 0.579 | 0.153 | 0.0008 | | | |
| | $NO_2^-$ | -1.619 | 0.660 | 0.0215 | | | |
| | $NO_3^-$ | -0.098 | 0.034 | 0.0086 | | | |
| Regeneration | T | 0.321 | 0.098 | 0.0031 | 0.695 | 12.8 | $1.42 \times 10^{-6}$ |
| | DO | 0.025 | 0.005 | 0.0003 | | | |
| | pH | -0.092 | 0.024 | 0.0008 | | | |
| | $NH_4^+$ | 0.386 | 0.126 | 0.0053 | | | |
| | $NO_3^-$ | -0.061 | 0.027 | 0.0340 | | | |
| Nitrification | $NO_2^-$ | 3.262 | 1.226 | 0.0165 | | | |
| $NH_4^+$ Ox. | $NO_2^-$ | 3.917 | 0.806 | 0.0001 | 0.496 | 23.6 | $7.35 \times 10^{-5}$ |
| $NO_2^-$ Ox. | Chl a | 0.655 | 0.275 | 0.0301 | 0.422 | 3.4 | 0.0202 |
| | $NH_4^+$ | 4.227 | 1.529 | 0.0138 | | | |



Figure 1

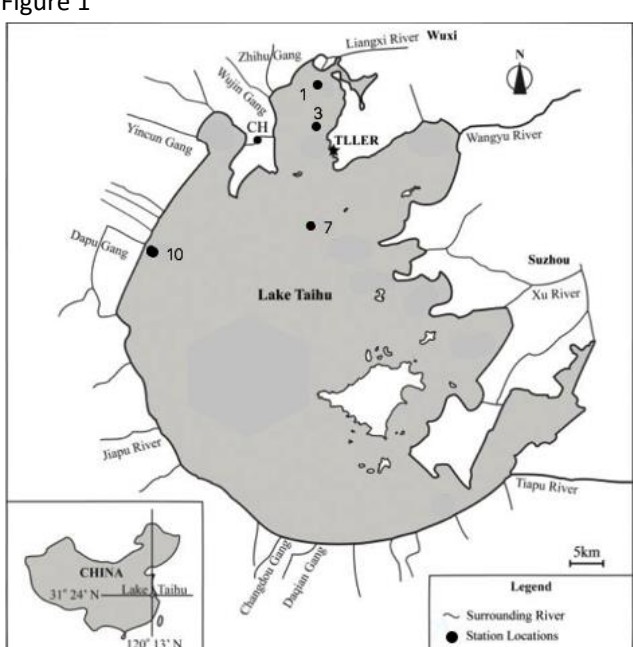





Figure 2

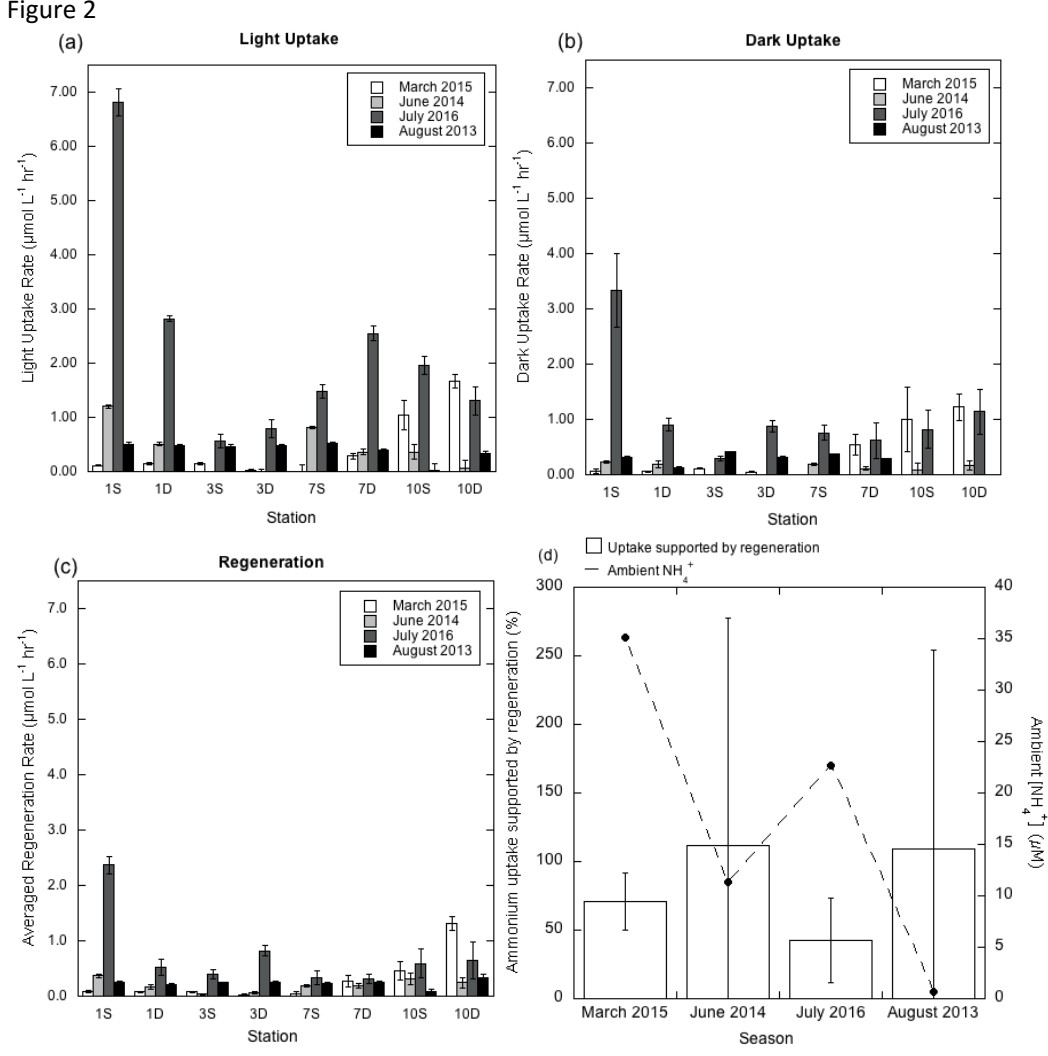





Figure 3

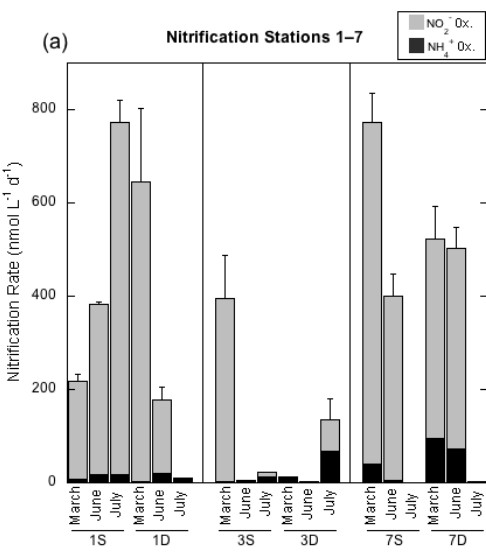

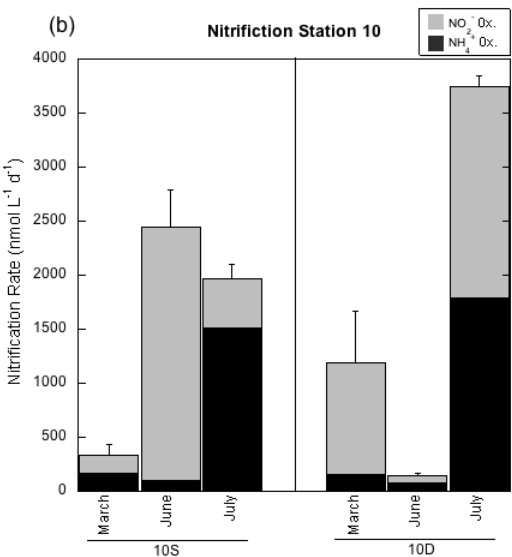




Figure 4

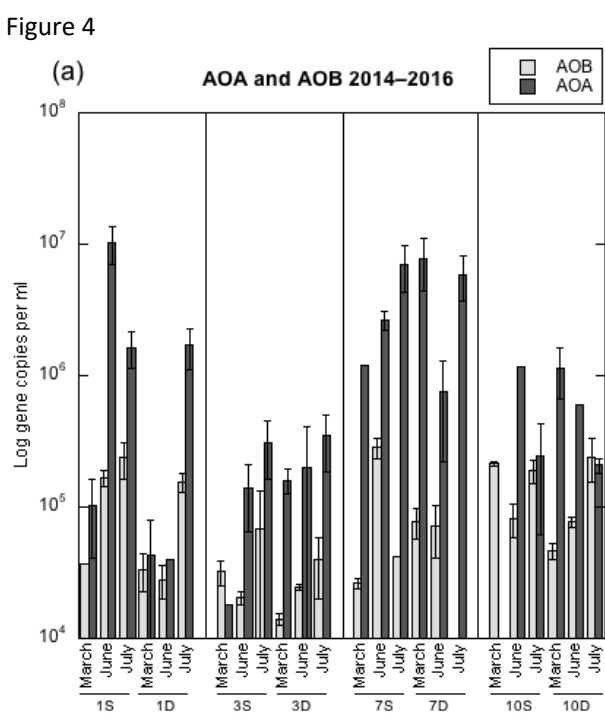

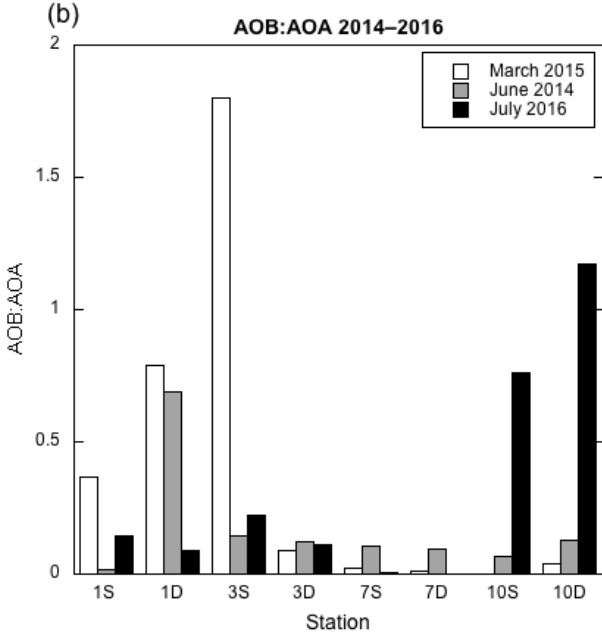