# Peer review of "Title: Nitrification and ammonium dynamics in Lake Taihu, China: seasonal competition"

_Biogeosciences, 2017_

## Referee Comment (RC1) · Anonymous Referee #1 · 26 Oct 2017

The authors present measurements of seasonal NH4 dynamics in Taihu Lake which is a hypertrophic lake in southeastern China. The lake has been the subject of many studies related to harmful cyanobacterial blooms. Here, Hampel et al. rely on established methods to evaluate N cycling in the lake over a cyanobacterial bloom cycle including a no bloom early spring sample. Specifically, the authors measure ammonia and nitrite oxidation and NH4 uptake and regeneration and couple these data to in situ physicochemical parameters and abundance of amoA (a proxy for ammonia oxidation). From the data, the authors calculate a regression model to describe N dynamics in the

lake. The authors conclude that fixed nitrogen concentrations (NH4, NO2, NO3) drive N dynamics and that these transformations are driven in part by seasonal changes in lake physicochemistry. The data presented indicate NH4 regeneration in the lake exceeds external N loadings. Thus, the authors posit that denitrification could efficiently remove N from the system if external N loading decreased and this would also help mitigate harmful blooms.

The authors present a well-cited manuscript with comprehensive methods and statistical treatment of the data. The data highlight important aspects of within system nitrogen cycling and N dynamics that are necessary to understand the formation and collapse of harmful blooms as well as their mitigation. Below I will detail some specific comments and questions that could potentially improve the manuscript:

1. Throughout the presentation, the authors indicate the Lake Taihu bloom is comprised of Microcystis spp.. However, blooms can be highly dynamic and vary in population structure. The study does not provide supporting data that bloom studied was indeed Microcystis spp.. Furthermore, Microcystis spp. can have highly plastic genomes leading to genomic rearrangements and perhaps altered physiology. As such, evidence that the Lake Taihu Microcystis populations resemble those for which Km values have been measured (i.e. Nicklisch and Kohl 1983; Baldia et al., 2007) should be provided.

2. The study aims to capture seasonal dynamics but the samples were collected from nonconsecutive time points (August 2013, June 2014, March 2015, July 2016). I realize that planning and coordinating field sampling campaigns can be challenging, but are there any supporting information that the conditions for these years are comparable, blooms are of similar magnitude and timing etc. There is also no evidence that the samples captured bloom samples. For instance, what is the evidence for an early summer bloom in June 2014 vs. the mid-summer bloom in July 2016. This relates to my comment above - can you provide evidence that, at least during your samples, the blooms were comprised of Microcystis spp. to support the conclusions about competition for NH4 between Microcystis and AOO.

3. The multiple regression model seems to convey what one would expect - N cycling dynamics are driven by N availability. While I appreciate the care taken to calculate the model, I'm not convinced it adds to the study as presented.

4. In the discussion, there is no consideration of other features of the lake, i.e. hydraulic residence, that may contribute to different rates of N cycling in the different zones. Because the lake is large with multiple inputs, including multiple inputs that presumably vary in levels of nutrient (N and P) delivered, this seems worth considering in interpreting the data.

Some more technical comments:

Line 101: maybe within the system, in situ (rather than internally)

Line 156: This is, in most cases, an elevated ammonium concentration than measured in situ. Can you justify the concentration added and expand more on the range used - i.e. did lower concentration samples receive less?

Line 223: targeting a region, or do these primers amplify the entire gene?

Results: The samples capture seasons over several years. Do you have regional precipitation and/or weather patterns to indicate that your samples are representative across this timescale?

Line 256, Table 1: Can you provide detection limits, especially for NH4+ concentrations that are below limit of detection (should be bdl in your table rather than 0.00).

Lines 321-322: Please provide the detection limits for qPCR (i.e. from you standard curves).

Lines 361: algal production or cyanobacterial?

Line 514: Or utilizing a different substrate? Is it possible that HAB Cyanos successfully compete for NH4+ but AOA can still use the urea - perhaps less efficiently. Its possible that there is no competition.

Line 526: See main comment #2 - It doesn't seem like the paper presents data on bloom formation and maintenance in Taihu. As such, I don't think this conclusion can be supported but your data. I suggest removing or presenting the speculative nature of this statements and other similar conclusions about bloom formation / progression.

Line 528: Assuming NH4 is the preferred substrate

––––––––––––––––––

---

## Referee Comment (RC2) · Anonymous Referee #2 · 31 Oct 2017

General: This paper presents interesting results on the connections between various processes of inorganic N cycling processes in a eutrophic shallow lake in China. Strong points concern the simultaneous measurements of uptake, nitrification and ammonium regeneration and the discussion on how these processes are connected one to another and what are the major controlling factors. Weaker points concern the heterogeneous discussion which sometimes get lost in small descriptive details such as comparison of rates between Taihu and other systems but without a real discussion on what explains differences, or differences between dark and light uptakes which have already largely

been presented in previous literature. I believe the paper would gain to be refocused on N-dynamics at ecosystem scale (which is already almost the case). Some clarification is also needed on the choice of the authors to present nitrification rates as the sum of ammonium oxidation and nitrite oxidation while the later one is one order of magnitude higher. How do you define nitrification? If it is the rate at which NH4 transforms to NO3 ultimately, then "total nitrification" should = ammonium oxidation, which is the rate limiting step. Or otherwise, then please justify.

Specific: L43: replace "N fixation" by "N2 fixation" L44: idem as L43 L57: maybe useful to say that NH4 can accumulate in systems when there is O2 limitation – which is relevant in eutrophic systems. L71: It might be useful to cite the role of O2 in this uncoupling: There are many "kinetic" studies that show that nitrite oxidation is more sensitive to low O2 levels than ammonium oxidation and that this causes the decoupling of both processes in many suboxic aquatic systems. See for example Guisasola et al 2005 and references therein. L90 and throughout the manuscript: the acronym cyanoHABs = cyonobacteria harmfull algal blooms is often use in the place of "cyanobacteria", as it is the here. Should be checked and corrected when needed. L91 Affinity for ammonium: needs a reference L96 the term potential uptake rate is a bit confusing as actually it is not an uptake rate but rather a consumption rate which includes ammonium oxidation... L149 atom% 15N: of ammonium? L166 idem L96 L170 Please explain here how you calculate "total nitrification" or "nitrification" and justify. See also comment on L306. L255 You forget Chla in this part of the results L256 Do the variables not vary spatially also? I think they do as you discuss the special situation of station 10 L272: early summer bloom – how do you know this is the early summer bloom? From chla? L277 early spring bloom: idem as L272 L282 summer bloom: idem as L272 L303 using other units for nitrification is confusing. I would recommend to have similar units especially as later on you consider the fraction of ammonium consumption due to nitrification. L306 How do you define the total nitrification rate? By definition nitrification is the 2 steps reaction NH4->NO2->NO3, as you said in the introduction, so it should be the rate at which NH4 is transformed into NO3. So as nitrite

oxidation is not the limiting step, it should correspond to ammonium oxidation. You use the sum I believe, which then represents the production of NO2+NO3? but then as most of the NO2 is not produced by NH4 oxidation (much slower rates) but comes from external inputs (or other process) it is not clear what this really represents ecologically? Please, this needs clarification. L329 in this discussion point it is not clear why you don't you calculate an integrated NH4 uptake rate per station taking into account light/dark rates and surface/depth rates? It would refocus this part of the discussion. Presenting distinct light and dark rates in the discussion is distracting from the major (and most interesting) points. L330-339 I do not see the use of comparing rates of Lake Taihu in such details with other lakes if there is no discussion on what might explain the differences – and I think it is not the topic of the paper to do so. This could be shorter and table 3 removed. L340 Replace "presumably due to photosynthetic phytoplankton activity" by "presumably due to reduced photosynthetic phytoplankton activity" L340-342 This statement needs a reference: ammonium uptake is by phototrophs is reduced in the dark, not blocked so I don't think you can extrapolate to saying that heterotrophs and chemolithotrophs dominate the uptake. You don't know. L344 ". . .which may have been due to higher precipitation and subsequent runoff; . . ." you mean more nutrient inputs? What about the phytoplankton biomass? do you have a bloom that might explain higher rates? I see a max in Chla indeed. And there is also plenty of nutrients. L355-358: proportion % cited here do not correspond to the values observed in figure 2d. L360: describing July as early summer is confusing as June could be early summer. . . Maybe just keep the months names L369 Why don't you do the same with uptake rates and nitrification? Would be interesting L388-390 "However, our results show that these external N loads are fueling high regeneration rates and suggest that microbial denitrification cannot keep pace with external N loads" I do not understand this. L394 which Nitrification are we dealing with here? Total? Ammonium oxidation? Nitrite oxidation? L394 "previously reported rates": were these rates measured the same way (as the sum of NH4 and NO2 oxidation)? This can make a big difference on reported rates. L402: nitrification or ammonium oxidation? L414-415 "Higher NO2

oxidation rates were expected, since NO3 is the product of NO2 oxidation, and NO2 oxidation relies on the product of NH4 oxidation" I don't understand this statement. NO2 oxidation also relies on external sources of NO2 to the lake. It is not clear how you can have 10 times higher NO2 oxidation compared to NH4 oxidation. Needs more clarification. L424 how do you calculate the contribution of nitrification to the uptake? Do you use NH4 oxidation? L451 idem L424 L454 but as NO2 oxidation rates are higher everywhere, bot steps are also uncoupled at the other stations of the lake. L505 AOB is an ammonium oxidizer so can only contribute to ammonium oxidation (not total as mentioned) L518 replace "driven by" by "correlated with". Being correlated do not mean they are "driven by"

---

## Author Comment (AC1) · 8 Dec 2017

Response to Referee # 1

1. Throughout the presentation, the authors indicate the Lake Taihu bloom is comprised of Microcystis spp.. However, blooms can be highly dynamic and vary in population structure. The study does not provide supporting data that bloom studied was indeed Microcystis spp.. Furthermore, Microcystis spp. can have highly plastic genomes leading to genomic rearrangements and perhaps altered physiology. As such, evidence that the Lake Taihu Microcystis populations resemble those for which Km values have been measured (i.e. Nicklisch and Kohl 1983; Baldia et al., 2007) should be provided.

*Thank you for pointing this out. Bloom composition was not part of our study, but published literature on cyanobacterial blooms in Taihu is extensive, and these blooms have always been associated with Microcystis (Chen et al., 2003; Otten and Paerl 2011; Tang et al., 2013 and 2014, and citations in the manuscript). Additionally, the years sampled in this study correspond to Microcystis identification in the following studies (Ma et al., 2016; Tang et al., 2017; Deng et al., 2014; Li et al., 2017; Su et al., 2017; Qian et al., 2017). We will add this information to our site description section of the methods:*
Historically, these blooms have been associated with toxin producing, non-$N_2$ fixing *Microcystis spp.,* which can form surface scums on the lake for up to 10 months per year (Chen et al., 2003; Duan et al., 2009; Ma et al., 2016; Otten and Paerl 2011).

Water samples were collected in August 2013, June 2014, March 2015, and July 2016. Each of these sampling events corresponded with a pronounced *Microcystis* bloom (Ma et al., 2016; Deng et al., 2014; Li et al., 2017; Su et al., 2017; Qian et al., 2017), except stations 7 and 10 in March 2015 (visual observation).

*The $K_m$ values from Nicklish and Kohl (1983) and Baldia et al., (2007) provided in the Discussion are meant to serve only as a reference to Microcystis studies done in culture. We did not perform any kinetic experiments in our study, but a kinetic study during a Microcystis bloom in Taihu showed extremely high $K_m$ values ranging from 76.9 to 112.9 μM (Yang et al., 2017). We will add the following text to the Discussion section:*
With a high saturation threshold and reported $K_m$ values from 26.5 μM to 37 μM (Baldia et al., 2007; Nicklisch and Kohl 1983) in culture and up to 112.9 μM in Taihu populations (Yang et al., 2017), Microcystis should be able to outcompete nitrifiers at the high ambient $NH_4^+$ concentrations in Taihu as nitrifiers may become saturated at much lower concentrations.

References not included in manuscript already:
Chen et al., 2003 https://doi.org/10.1093/plankt/25.4.445
Deng et al., 2014 doi: 10.1111/fwb.12330
Li et al., 2017 http://dx.doi.org/10.1016/j.jglr.2017.04.005
Otten and Paerl 2011 doi:10.1007/s00248-011-9884-x
Su et al., 2017 http://dx.doi.org/10.1016/j.hal.2017.08.007
Qian et al., 2017 doi: 10.1007/s00128-017-2149-8
Yang et al., 2017 doi:10.1016/j.hal.2017.04.001

2. The study aims to capture seasonal dynamics but the samples were collected from nonconsecutive time points (August 2013, June 2014, March 2015, July 2016). I realize that planning and coordinating field sampling campaigns can be challenging, but are there any supporting information that the conditions for these years are comparable,

blooms are of similar magnitude and timing etc. There is also no evidence that the samples captured bloom samples. For instance, what is the evidence for an early summer bloom in June 2014 vs. the mid-summer bloom in July 2016. This relates to my comment above - can you provide evidence that, at least during your samples, the blooms were comprised of Microcystis spp. to support the conclusions about competition for NH4 between Microcystis and AOO.

*Historically, blooms in Taihu have occurred 8–10 months of the year (Ma et al., 2016). A year long study in 2014 showed that blooms usually start in March/April and last until December (Li et al. 2017). Our chlorophyll data (Table 1) shows the seasonal bloom characteristics, with lowest values in March and highest in June/August.*

*We will add a paragraph to the results section discussing the chlorophyll data in more detail:*
Chlorophyll a data showed seasonal variation. Overall, lowest values were recorded in March 2015 (mean = 11.1 µg L$^{-1}$), but bloom conditions (> 20µg L$^{-1}$; Xu et al., 2015) were observed at some locations (20.3 µg L$^{-1}$ at station 3, and visual observations at Station 1 and several other areas of the lake). Bloom conditions were also observed in June 2014 (mean = 36.6 µg L$^{-1}$), July 2016 (mean = 58.1 µg L$^{-1}$), and August 2013 (43.7 µg L$^{-1}$).

*We will also provide a supplemental table of field notes and observations describing bloom status during each sampling event.*
Reference:
Xu et al., 2015 doi:10.1021/es503744q

3. The multiple regression model seems to convey what one would expect - N cycling dynamics are driven by N availability. While I appreciate the care taken to calculate the model, I'm not convinced it adds to the study as presented.

*The purpose of the multiple regression model was to show that NH$_4$$^+$ dynamics correlate with seasonally variable temperature and DO. While this relationship is not present in the Kendall's correlation table, a more complex model in Table 4 shows the importance of multiple variables at the same time. We thought it was important, for a complex environment like Taihu, to show that temperature plays a role in NH$_4$$^+$ dynamics. We pointed this out in lines 518–523. The model also highlights the lack of temperature relationship with nitrification rates, which was an unexpected result.*

4. In the discussion, there is no consideration of other features of the lake, i.e. hydraulic residence, that may contribute to different rates of N cycling in the different zones. Because the lake is large with multiple inputs, including multiple inputs that presumably vary in levels of nutrient (N and P) delivered, this seems worth considering in interpreting the data.

*This is an excellent point. However, an in-depth discussion of these physical factors is beyond the scope of the paper. We are part of a larger collaborative project, and data from this study will be incorporated into a Lake Taihu ecosystem model by the Hellweger lab at Northeastern University. This model will include features like N and P loading, hydraulic residence time, N cycling rates, chlorophyll dynamics, etc.*

*We agree that there should be a general discussion of the residence time, however. We will add this information to our methods section:*
Taihu Lake has a relatively long residence of approximately 280–300 days (Paerl et al., 2014; Xu et al., 2010).

Some more technical comments:
Line 101: maybe within the system, in situ (rather than internally)
*We will revise to "within the system"*

Line 156: This is, in most cases, an elevated ammonium concentration than measured in situ. Can you justify the concentration added and expand more on the range used - i.e. did lower concentration samples receive less?

*The goal of the substrate additions in these uptake/regeneration experiments was to add more-than-trace levels to ensure that all of the label was not taken up during the incubations. The substrate additions depended in part on bloom status, and our goal was to add the label concentration at an equivalent value to the most recent monitoring data we could obtain for NH4 concentrations, or at least 8 uM (even when concentrations are low, recycling rates can be quite high). Lower level additions coincided with low ambient concentrations and lighter blooms, while higher level additions were conducted at sites with heavy blooms and/or high ambient NH4 concentrations. We will add text in the methods to clarify our isotope labelling approach. We also point out that the uptake rate is considered potential (line 166), because these substrate additions exceed ambient levels. However, the regeneration rates are actual, not potential.*

Line 223: targeting a region, or do these primers amplify the entire gene?
*Targeting a region of the amoA gene, as stated in the manuscript (line 223).*

Results: The samples capture seasons over several years. Do you have regional precipitation and/or weather patterns to indicate that your samples are representative across this timescale?
*Our sampling dates were representative of seasonal conditions in the region, specific to this subtropical climate zone. Our sampling events did not correspond with any extreme weather patterns (e.g., typhoons, droughts). Temperature and precipitation patterns were average for this climate region and are available online.
(https://www.wunderground.com/history/airport/ZSSS/2013/8/1/DailyHistory.html).
We added this sentence to the Methods:*
Our sampling dates were representative of seasonal conditions in the region, specific to this subtropical climate zone and did not correspond with any extreme weather patterns (e.g., typhoons, droughts). Temperature and precipitation patterns were average for this climate region.

Line 256, Table 1: Can you provide detection limits, especially for NH4+ concentrations that are below limit of detection (should be bdl in your table rather than 0.00).
*Thank you for pointing this out. We will change 0.00 values to BDL and provide a footnote with the table stating detection limits for our nutrient analyses:*
*Nutrient analysis detection limits: $NH_4^+$ = 0.04 µM; $NO_x$ = 0.04 µM; OP = 0.008 µM.

Lines 321-322: Please provide the detection limits for qPCR (i.e. from you standard curves).
*The detection limit for was 980 copies/ml for AOB and 4807 copies/ml for AOA, calculated from standard deviation of the lowest standard multiplied by student t test value. These calculated detection limits do not represent the greatest sensitivity possible with our method, as the*

*standard concentrations were selected to bracket the expected environmental concentrations. Indeed, our reported values are above the detection limit for both AOA (by two orders of magnitude) and AOB.*

Lines 361: algal production or cyanobacterial?
*Algal production. We can't rule out other organisms.*

Line 514: Or utilizing a different substrate? Is it possible that HAB Cyanos successfully compete for NH4+ but AOA can still use the urea - perhaps less efficiently. Its possible that there is no competition.
*As stated in the manuscript (lines 512–514), we consider the possibility of AOA utilizing urea and/or cyanate; however, these substrates might be used less efficiently, and at much lower rates than $NH_4^+$. Palatinszky et al. (2015) shows lower protein concentration in AOA grown on cyanate than on $NH_4^+$. Tolar et al. (2016) shows lower oxidation rates of urea than $NH_4^+$ in samples from coastal Georgia and South Atlantic, concluding that urea-derived N does not play a major role in temperate regions.*
*We will add text as appropriate to clarify this:*
Recent studies show that AOA can oxidize cyanate (Palatinszky et al., 2015) and urea (Tolar et al., 2016), although growth and oxidation rates may be slower. Therefore, it is possible that AOA might be playing a different an expanded role in Taihu beyond just ammonia oxidation.

Line 526: See main comment #2 - It doesn't seem like the paper presents data on bloom formation and maintenance in Taihu. As such, I don't think this conclusion can be supported but your data. I suggest removing or presenting the speculative nature of this statements and other similar conclusions about bloom formation / progression.
*We will revise this sentence to:*
This study highlights the importance of water column $NH_4^+$ regeneration in providing a large proportion of the substrate necessary to sustain cyanoHABs. The results also show that nitrification does not account for a large proportion of $NH_4^+$ demand during cyanobacterial blooms in Taihu.

Line 528: Assuming NH4 is the preferred substrate
*Many studies have shown that $NH_4^+$ is the preferred N source for cyanobacteria, especially non-N-fixers like Microcystis (Blomqvist et al., 1994; Glibert et al., 2015; Gobler et al., 2016; McCarthy et al., 2009; lines 52–56).*
Gobler et al., 2016 http://dx.doi.org/10.1016/j.hal.2016.01.010

---

## Author Comment (AC2) · 8 Dec 2017

Response to Referee # 2

How do you define nitrification? If it is the rate at which NH4 transforms to NO3 ultimately, then "total nitrification" should = ammonium oxidation, which is the rate limiting step. Or otherwise, then please justify.

*The reviewer makes a good point that these terms need clarification. It is true that produced $^{15}NO_3^-$ must have originated from $^{15}NH_4^+$ and that ammonia oxidation is usually the rate limiting step in nitrification, making total $NO_3^-$ production inclusive of ammonia oxidation. We will rephrase the definition of nitrification (focused on total nitrification; e.g. the sum of $NO_2^-$ and $NO_3^-$ production), as well as our discussion of the partitioning of the product of $^{15}NH_4^+$ additions (i.e., comparing $^{15}NO_2^-$ to the $^{15}NO_3^-$ pool). We will revise the text in the methods, results, and discussion accordingly. For example, we will review this issue in the Discussion as follows:*

Nitrification rates were positively correlated with ambient $NH_4^+$, $NO_2^-$, and $NO_3^-$ concentrations ($p < 0.05$; Table 2), as expected. Substrate concentrations drive $NH_4^+$ oxidation rates and, therefore, end-product pools, since it is the rate limiting step of nitrification (i.e., completion of nitrification is dependent on the first step). Accumulation of $^{15}NO_3^-$ exceeded accumulation of $^{15}NO_2^-$ by a factor of 9 at stations 1, 3, and 7, across all sampling events (Fig. 3a), indicating that $NO_2^-$ oxidation is keeping pace with or exceeding $NH_4^+$ oxidation. Higher accumulation of $^{15}NO_3^-$ was expected, since $NO_3^-$ is the final product of total nitrification.

In contrast, at station 10, while accumulation of $^{15}NO_3^-$ exceeded $^{15}NO_2^-$ in March 2015 and June 2014, in July 2016 accumulation of $^{15}NO_2^-$ was three times higher in surface water and comparable at depth (Fig. 3b). Additionally, there was a significant pool of nitrite: ambient $NO_2^-$ concentration at station 10 in July 2016 ranged from 9.6 µM in surface water to 8.4 µM at depth (Table 1). This accumulation of $NO_2^-$ could indicate that $NO_2^-$ oxidizers were saturated, as reported 418 Km values for $NO_2^-$ oxidation in an oligotrophic, oxygen deficient region in the ocean were $0.25 \pm 0.16$ µM (Sun et al., 2017). However, culture experiments report $K_m$ values ranging from 6–544 µM for *Nitrospira*, *Nitrobacter*, and *Nitrotoga* spp. (Blackburne et al., 2007; Nowka et al., 2015; Ushiki et al., 2017).

Specific: L43: replace "N fixation" by "N2 fixation"
*We will revise to $N_2$ fixation.*

L44: idem as L43 L57: maybe useful to say that NH4 can accumulate in systems when there is O2 limitation – which is relevant in eutrophic systems.

*This is true in many eutrophic systems, but Lake Taihu is very shallow (2 m on average) and well-mixed. We do not observe $O_2$ depletion at depth, and stratification is rare (Qin et al., 2004); therefore, we do not expect $O_2$ limitation to play a role in N cycling processes in the well-mixed water column.*

Reference:
Qin, B., Hu, W., Gao, G., Luo, L. and Zhang, J., 2004. Dynamics of sediment resuspension and the conceptual schema of nutrient release in the large shallow Lake Taihu, China. *Chinese Science Bulletin*, *49*(1), pp.54-64.

L71: It might be useful to cite the role of O2 in this uncoupling: There are many "kinetic" studies that show that nitrite oxidation is more sensitive to low O2 levels than ammonium oxidation and that this causes the decoupling of both processes in many suboxic aquatic systems. See for example Guisasola et al 2005 and references therein.

*Indeed, this is true, and thank you for the reference. As stated above, however, Taihu is well-mixed, very shallow, and not susceptible to suboxic conditions in the water column.*

L90 and throughout the manuscript: the acronym cyanoHABs= cyonobacteria harmfull algal blooms is often use in the place of "cyanobacteria",as it is the here. Should be checked and corrected when needed.
*Thank you for pointing this out. We will correct the inconsistent use of "cyanobacteria" and "cyanoHABs".*

L91 Affinity for ammonium: needs a reference
*We will add appropriate references here, such as Martens-Habbena et al., (2009) and Baldia et al., (2007).*

L96 the term potential uptake rate is a bit confusing
as actually it is not an uptake rate but rather a consumption rate which includes ammonium oxidation
*We agree that our wording here is confusing. We will rephrase for clarity:*
We measured community $NH_4^+$ uptake and regeneration rates, as well as nitrification rates, under different bloom conditions to help determine how cyanoHABs influence $NH_4^+$ fluxes.

*L149 atom% 15N: of ammonium?*
*Yes, we will clarify in the revised manuscript that we mean atom % of 15N-$NH_4^+$*

*L166 idem L96 L170 Please explain here how you calculate "total nitrification" or "nitrification" and justify. See also comment on L306.*
*We agree with the reviewer's comments, and we will modify the text per the example above.*

L255 You forget Chla in this part of the results
*Thank you for pointing this out. We will add chlorophyll results to the results section:*
Chlorophyll a data showed seasonal variation. Overall, lowest values were recorded in March 2015 (mean = 11.1 µg L$^{-1}$), but bloom conditions (> 20µg L$^{-1}$; Xu et al., 2015) were observed at some locations (20.3 µg L$^{-1}$ at station 3, and visual observations at Station 1 and several other areas of the lake). Bloom conditions were also observed in June 2014 (mean = 36.6 µg L$^{-1}$), July 2016 (mean = 58.1 µg L$^{-1}$), and August 2013 (43.7 µg L$^{-1}$).
Reference:
Xu et al., 2015 doi:10.1021/es503744q

L256 Do the variables not vary spatially also? I think they do as you discuss the special situation of station 10
*Yes, we will add "spatially" to the sentence.*

L272: early summer bloom – how do you know this is the early summer bloom? From chla?
L277 early spring bloom: idem as L272 L282 summer bloom: idem as L272.
*For clarification, we removed all "early bloom", "mid bloom", and "late bloom" descriptions. Instead, we kept the month names only.*

L303 using other units for nitrification is confusing. I would recommend to have similar units especially as later on you consider the fraction of ammonium consumption due to nitrification.
*We aimed to report total nitrification rates in units consistent with the majority of the literature. $nmol\ L^{-1}\ d^{-1}$ is usually used for these rates (Bristow et al., 2015; Heiss and Fulweiler 2016; Newell et al. 2011; Ward and Kilpatrick 1991). Uptake and regenerations rates are much faster, and are on a micromolar scale (also a standard literature unit for these rates; Gardner et al., 2001; James et al., 2011; McCarthy et al., 2013). For ease of unit conversion/comparison, we will add an additional axis to Fig. 3 to show µM/hr units.*

L306 How do you define the total nitrification rate? By definition nitrification is the 2 steps reaction NH4->NO2->NO3, as you said in the introduction, so it should be the rate at which NH4 is transformed into NO3. So as nitrite
oxidation is not the limiting step, it should correspond to ammonium oxidation. You use the sum I believe, which then represents the production of NO2+NO3? but then as most of the NO2 is not produced by NH4 oxidation (much slower rates) but comes from external inputs (or other process) it is not clear what this really represents ecologically? Please, this needs clarification.
*In environmental studies, it is not uncommon for nitrite oxidation to exceed ammonia oxidation (Bristow et al., 2015; Clark et al., 2008; Fussel et al., 2012; Heiss and Fulweiler 2016; Ward and Kilpatrick 1991). We agree that the way we reported the rates is confusing; therefore, we will revise the manuscript as stated above. We will rephrase the definition of nitrification (focused on total nitrification; e.g. the sum of $^{15}NO_2^-$ and $^{15}NO_3^-$ production), as well as our discussion of the partitioning of the product of $^{15}NH_4^+$ additions (i.e., comparing $^{15}NO_2^-$ to the $^{15}NO_3^-$ pool).*
Reference:
Clark et al., 2008 doi:10.2307/40006149
Ward and Kilpatrick 1991 https://doi.org/10.1016/0278-4343(90)90016-F

L329 in this discussion point it is not clear why you don't you calculate an integrated NH4 uptake rate per station taking into account light/dark rates and surface/depth rates? It would refocus this part of the discussion. Presenting distinct light and dark rates in the discussion is distracting from the major (and most interesting) points.
*We did not integrate light and dark rates so that we could highlight and distinguish the differences between total community uptake (light) from non-photoautotrophic uptake (dark). We think this is an important part of the study. We did not integrate the surface and deep rates because the system is shallow and well-mixed, and Microcystis can regulate its buoyancy to form surface scums. Additionally, without high-resolution depth profiles of relevant physicochemical parameters, it is difficult to distinguish differences in surface and bottom water masses.*

L330-339 I do not see the use of comparing rates of Lake Taihu in such details with other lakes if there is no discussion on what might explain the differences – and I think it is not the topic of the paper to do so. This could be shorter and table 3 removed.
*We agree that our presentation of the system comparisons can be improved. We wanted to include the table to give an overview of uptake and regeneration rates in other freshwater systems, and also to highlight differences in rates relative to trophic status (e.g., eutrophic vs hypereutrophic). We will add chlorophyll a values as an indicator of trophic status in the table and clarify this in the text:*

While potential $NH_4^+$ uptake rates increase with chlorophyll a ($p < 0.05$), the relative proportion of community uptake that can be supported by regeneration remains consistent (Table 3).

L340 Replace "presumably due to photosynthetic phytoplankton activity" by "presumably due to reduced photosynthetic phytoplankton activity"
*Thank you for pointing this out. We will revise as suggested.*

L340-342 This statement needs a reference: ammonium uptake is by phototrophs is reduced in the dark, not blocked so I don't think you can extrapolate to saying that heterotrophs and chemolithotrophs dominate the uptake. You don't know.
*Phototrophs usually take up nutrients in the dark when they are nutrient limited (Cochlan et al., 1991). Taihu is generally nutrient replete, so we speculate that the dark uptake can be mostly attributed to heterotrophs and chemolithoautotrophs. We will revise these two sentences to clarify this information:*
"Photoautotrophs may continue to assimilate nutrients in the dark under nutrient limitation (Cochlan et al., 1991). However, Taihu is nutrient replete, so dark uptake rates can likely be attributed to heterotrophic or chemolithoautotrophic organisms."

L344 "which may have been due to higher precipitation and subsequent runoff" you mean more nutrient inputs? What about the phytoplankton biomass? do you have a bloom that might explain higher rates? I see a max in Chla indeed. And there is also plenty of nutrients.
*Yes, more runoff = more nutrient inputs.*
*There is strong bloom evidence looking at July 2016 chlorophyll (above the chla threshold of 20 $\mu g\ L^{-1}$; Xu et al., 2015) and high nutrient concentrations (Table 1). We will add a Supplementary table with our field notes and visual observations to help clarify this point.*

L355-358: proportion % cited here do not correspond to the values observed in figure 2d.
*Thank you for pointing this out. We accidentally uploaded an outdated graph. Here is the correct version that corresponds to the values in the text and will be included in the revised manuscript.*
Figure 2d

[Figure]

L360: describing July as early summer is confusing as June could be early summer: : : Maybe just keep the months names
*Thank you for pointing this out. We kept only the month names for clarification.*

L369 Why don't you do the same with uptake rates and nitrification? Would be interesting.
*The purpose of this extrapolation was to compare external N loading to $NH_4^+$ provided by regeneration. We think that it is an important highlight of this paper. We will add a sentence comparing the extrapolated uptake rates to total nitrogen load. There are high standing pools of $NH_4^+$, $NO_2^-$ and $NO_3^-$ and cycling rates are high; therefore, a nitrification extrapolation would not be informative.*

L388-390 "However, our results show that these external N loads are fueling high regeneration rates and suggest that microbial denitrification cannot keep pace with external N loads" I do not understand this.
*We agree that this sentence is confusing. We split it into two sentences for clarification:*
However, our results show that these external N loads lead to higher biomass and fuel high regeneration rates. Combined with high ambient nutrient concentrations, these results suggest that microbial denitrification cannot remove enough N to effectively mitigate the high external N loading.

L394 which Nitrification are we dealing with here? Total? Ammonium oxidation? Nitrite oxidation?
*We will clarify that we are talking about total nitrification rates here.*

L394 "previously reported rates": were these rates measured the same way (as the sum of NH4 and NO2 oxidation)? This can make a big difference on reported rates.

*Rates reported in Lake Okeechobee were measured using a $^{15}NO_3^-$ pool dilution method. Rates in Lakes Superior and Mono (Line 397), however, were measured using the same $^{15}NH_4^+$ tracer addition technique. Rates in Lake Mendota and the Paerl River Estuary were not measured using $^{15}N$ stable isotope methods. We will add this information to the text:*

Published nitrification rates in lakes include the water columns of saline Lake Mono, CA (60–480 nmol $L^{-1}$ $d^{-1}$; Carini and Joye, 2008) and Lake Superior, USA (0–51 nmol $L^{-1}$ $d^{-1}$; Small et al., 2013), both measured via $^{15}NH_4^+$ tracer additions, and Lake Okeechobee, FL (67–97 nmol $L^{-1}$ $hr^{-1}$; James et al., 2011) measured via the $^{15}NO_3^-$ pool dilution method (Carini et al. 2010). Rates on this scale were previously reported only in eutrophic Lake Mendota (WI; 1700 – 26000 nmol $L^{-1}$ $hr^{-1}$; Hall, 1986) and the Paerl River Estuary (China; 2100 – 65100 µmol $L^{-1}$ $d^{-1}$; Dai et al., 2008). However, these rates were measured from accumulation of $NO_2^-$ and $NO_3^-$, not stable isotope additions.

L402: nitrification or ammonium oxidation?
*We will clarify in the revision that we are talking about total nitrification rates.*

L414-415 "Higher NO2 oxidation rates were expected, since NO3 is the product of NO2 oxidation, and NO2 oxidation relies on the product of NH4 oxidation" I don't understand this statement. NO2 oxidation also relies on external sources of NO2 to the lake. It is not clear how you can have 10 times higher NO2 oxidation compared to NH4 oxidation. Needs more clarification.
*We will make necessary changes as stated above.*

L424 how do you calculate the contribution of nitrification to the uptake?
Do you use NH4 oxidation?
*We use total nitrification to determine the contribution of nitrification to total $NH_4^+$ uptake.*

L451 idem L424 L454 but as NO2 oxidation rates are higher everywhere, bot steps are also uncoupled at the other stations of the lake.
*We calculated the contribution of nitrification to uptake from total nitrification rates.*

L505AOB is an ammonium oxidizer so can only contribute to ammonium oxidation (not total as mentioned)
*In this case, total nitrification originated from $^{15}NH_4^+$.*

L518 replace "driven by" by "correlated with". Being correlated do not mean they are "driven by"
*Good point. We will change "driven by" to "correlated with"*